

# Investigating the spatiotemporal features of glacier elevation changes over the southeastern Tibetan Plateau using multisource satellite data

Xin Luo[1], Hongping Zeng[2], Zhen Ye[3]

[1] School of Earth Sciences, Yunnan University, Kunming, 650500, China

[2] Institute of International Rivers and Eco-security, Yunnan University, Kunming, 650500, China

[3] College of Surveying and Geoinformatics, Tongji University, Shanghai, 200092, China

*Correspondence to*: Xin Luo (xinluo_xin@ynu.edu.cn)

**ABSTRACT**. Glaciers in the southeastern Tibetan Plateau (SETP) feature the largest maritime glaciers on the Tibetan Plateau (TP) and have experienced accelerated melting in recent decades. Investigating the spatiotemporal features of glacier elevation changes in the SETP remains a challenging task since this region is highly heterogeneous and high spatiotemporal resolution observations for region-wide glacier change measurements are still insufficient. To better understand the spatiotemporal variations in glacier elevation changes in the SETP, multisource satellite observations, including ASTER DEM, ICESat, ICESat-2 and CryoSat-2, are integrated in this study. We derive the spatially resolved glacier change for each year based on the $0.5° \times 0.5°$ geographical tiles, and the obtained glacier elevation change rate of the entire SETP is $-0.710 \pm 0.046 \, m/yr$ during 2000–2022. We divided the study period into a recent decade and the previous decade and found that glacier thinning accelerated at a rate of 31.2% in the recent decade. We evaluated the correlation between the elevation measurements of different satellites and found that the elevation measurement of ICESat-2 had a slight negative bias relative to the measurements of the other satellites. The ICESat-2 elevation measurements of the strong beam and weak beam were also compared, and no significant difference was observed. We also compared the CryoSat-2 swath measurements with the Level-2 (L2) measurements, and we found that the CyroSat-2 swath data agreed significantly more with the other satellite data than with the L2 measurements. A comprehensive comparison is carried out for the glacier elevation changes obtained in existing studies. Our estimates are highly consistent with those of new published studies and have a finer temporal scale and less estimation uncertainty.



**Keywords**: Southeastern Tibetan Plateau, glacier elevation change, ASTER stereo images, ICESAT data,
CryoSat-2.

## 1    Introduction

Glaciers are sensitive to climate change (Bach et al., 2018; Yao et al., 2012), and most of the world's
glaciers, excluding the Greenland and Antarctic ice sheets, have experienced accelerated ice melting and
mass loss in recent decades (Hugonnet et al., 2021; IPCC, 2020). As one of the most climate-sensitive
constituents of the natural landscape, glacier changes have raised broad concerns, such as contributions
to river and sea-level rise changes (Gardner et al., 2013; Jacob et al., 2012), sustainable water resource
supplies for downstream humans (Milner et al., 2017), and hazards such as glacier lake outburst floods
(Bolch et al., 2008; Carrivick and Tweed, 2016).
The High Mountain Asia (HMA) region hosts the largest glacier concentration outside the polar regions,
and these glaciers prominently contribute to streamflow in one of the most populated areas of the world
(Brun et al., 2017). Current studies have revealed that the glaciers in HMA are greatly retreating (Brun
et al., 2017; Gardner et al., 2013; Yao et al., 2012). Brun et al. (2017) illustrated that the glacier elevation
change rate for the total HMA region was -0.21 $\pm$ 0.05 m/yr during 2000-2016. Shean et al. (2020)
revealed that the total HMA glacier mass change rate during 2000-2018 was $-0.19 \pm 0.03$ m w.e./yr.
Glacier elevation change shows large spatial variability in HMA; that is, Shen et al. (2020) illustrated a
positive glacier elevation change rate of $0.047 \pm 0.06$ m/yr in Kunlun for 2000-2018 (Shean et al.,
2020). Kääb et al. (2015) illustrated that a slight negative glacier elevation change rate of -0.1$\pm$ 0.06 m/yr
was observed in Karakoram for 2003-2008 (Kääb et al., 2015), and Brun et al. (2017) illustrated that the
most negative elevation change rate of -0.73 $\pm$ 0.27 m/yr was found in Nyainqentanglha for 2000-2016.
Among the subregions of the HMA region, the SETP has experienced dramatic glacier mass loss over
the past two decades; however, the glacier elevation changes in the SETP have not been well quantified
by existing studies due to inconsistent results. For example, the Nyainqentanglha region (the main SETP)
was reported to experience a very negative elevation change rate of -1.34 $\pm$ 0.29 m yr[1] from 2003–2008
according to Kääb et al. (2015), while a slight negative change rate of approximately -0.30 $\pm$ 0.13 m/yr[1]
from 2003–2008 was revealed by Gardner et al. (2013). According to in situ and remotely sensed



observations, the glaciers in the SETP region have experienced the strongest recession in HMA since the
2000s, and this recession in the SETP accounts for approximately one-quarter to one-third of the total
glacier mass loss in HMA (Brun et al., 2017; Kääb et al., 2015; Neckel et al., 2014). Glacier elevation
changes in the SETP have drawn the most attention from glaciologists, climatologists, and geoscientists,
and great efforts have been made to improve elevation change estimations (Brun et al., 2017; Gardner et
al., 2013; Shean et al., 2020). However, estimates of glacier elevation changes are still limited in accuracy
and are inconsistent with each other for the SETP region. For example, various estimates of $-0.40 \pm$
$0.41$ m/yr for 2003–2009 (Gardner et al., 2013), $-0.81 \pm 0.32$ m/yr for 2003–2008 (Neckel et al., 2014),
$-0.68 \pm 0.22$ m/yr for 2000–2016 (Brun et al., 2017), $-1.11 \pm 0.11$ m/yr for 2010–2019 (Jakob et al.,
2021), $-0.54 \pm 0.16$ m/yr for 2000–2018 (Shean et al., 2020), $-0.69 \pm 0.35$ m/yr for 2000–2019
(Hugonnet et al., 2021), and $-0.73 \pm 0.18$ m/yr for 2003–2020 (Zhao et al., 2022) have been reported in
existing studies.
Glacier observations are mainly obtained through spaceborne remote sensing. There are four major
approaches for observing glaciers: 1) Satellite images: These images have long been used for glacier
observation and have provided abundant historical glacier change information along with climate change
information (Ke et al., 2015). Mountain glaciers can typically be automatically or semiautomatically
delineated with satellite images, such as Landsat, Advanced Spaceborne Thermal Emission and
Reflection Radiometer (ASTER), Sentinel-2, or Sentinel-1 images (Pope and Rees, 2014, Robson et al.,
2020; Zhou and Zhen, 2017; Peng et al., 2023). Satellite-based optical stereo images, such as ASTER
stereo images, WorldView stereo image, and Corona KH-4A stereo images, can be used to determine
glacier elevations (Shean et al., 2020; Wang et al., 2021; Muhammad and Tian, 2020; Zhao et al., 2022;
King et al., 2023; Piermattei et al., 2023; Ghuffar et al., 2022). 2) Geodetic digital elevation model (DEM)
differencing: DEM differencing is the most common approach for calculating the elevation change in an
entire glacier area and enables the estimation of glacier mass change over time (Cogley et al., 2011;
Robson et al., 2022). Generally, spaceborne optical and radar sensors are two main sources for producing
DEMs for calculating elevation changes in large-scale glacierized regions. Spaceborne radar DEMs, such
as those from the Shuttle Rader Topography Mission (SRTM) C-band data and TanDEM X-band data
(Ke et al., 2020; Guan et al., 2022), are generated using interferometric synthetic aperture radar (InSAR).





The radar signal penetrates snow and ice, and the penetration depth is a matter of debate and leads to
uncertainty in glacier elevation estimation (Barundun et al., 2015; Kääb et al., 2015). Spaceborne optical
digital models (DEMs) are generated using stereo images (Bhushan et al., 2021; King et al., 2023;); the
generated DEMs are free of ice/snow penetration, while optical stereo images are unavailable in cloudy
regions. 3) Satellite altimetry. The Ice, Cloud, and Land Elevation Satellite (ICESat) missions from 2003
to 2009 and the following ICESat-2 mission launched in 2018 have created nearly global datasets that
have been widely used for glacier elevation change estimation (Zwally et al., 2002; Smith et al., 2020).
Additionally, the repeat cycle of CryoSat-2 has recently been exploited for glacier elevation change
monitoring (Zhao et al., 2022). Satellite altimetry has been an efficient approach for investigating glacier
elevation changes due to the continuous temporal observation of the Earth's surface; however, this
approach is subject to sparse spatial sampling at low latitudes, and the precision of measurement is
usually limited in complex terrains (Treichler et al., 2016). 4) Satellite gravity. The Gravity Recovery
and Climate Experiment (GRACE) satellites can be used to detect glacier change-derived variations in
Earth's gravity field. The GRACE estimates have very coarse spatial resolution (~150 km) and are
usually contaminated by water storage changes in adjacent regions; therefore, hydrological observation
is needed to cooperate with the determination of glacier changes when using satellite gravity (Chen et
al., 2017; Yi et al., 2020). Above all, a variety of data have been explored for glacier elevation changes
monitoring, and different data sources have respective advantages and limitations; therefore, recent
studies have explored the integration of multisource data to improve the precision of glacier monitoring
(Zhao et al., 2022; Ke et al., 2020; Muhammad and Tian, 2020; Bhattacharya et al., 2023).
Existing studies tend to focus on large-scale glacier elevation change monitoring, e.g., for the whole
HMA region or at the global scale (Brun et al., 2017; Hugonnet et al., 2021). These studies revealed that
the SETP has undergone accelerated glacier melting in recent years; however, the detailed spatiotemporal
variability in the glacier change rate remains poorly understood. In this study, we integrated multisource
satellite data to enhance the estimate of glacier elevation change in the SETP during the period 2000–
2022. Specifically, ASTER stereo images were used to generate time-series DEM data, and the generated
DEMs were then combined with altimetry data (including ICESat, ICESat-2 and CryoSat-2) to
investigate the spatiotemporal features of glacier elevation changes in the SETP. The major methods of



this study included the following steps: 1) We fully exploited the ASTER stereo image, ICESat, ICESat-
2 and CryoSat-2 altimetry data to improve the estimation of glacier elevation change. 2) We investigated
the spatiotemporal features of glacier elevation changes in the SETP and better quantified the
heterogeneity of glacier elevation changes in this region. 3) Comprehensive cross-analysis was
performed for elevation measurements taken by multisource satellites, as well as estimates of glacier
elevation changes obtained from existing studies. The remainder of this paper is structured as follows.
Section 2 introduces the study area and dataset. Section 3 describes the methodology. Section 4 provides
the results and illustrates the variations in glacier elevation change in the SETP. In Section 5, elevation
measurements from different satellites and estimates of glacier elevation changes from existing studies
are discussed. Finally, we draw conclusions in Section 6.
**2    Study area and data**
**2.1 Study area**
The SETP is a broad mountainous area that covers the Western Hengduan Mountains, the Eastern
Nyainqentanglha Ranges, and the Eastern Himalayas; the SETP boundary defined by Zhao et al. (2022)
was used in our study (Fig. 1). This region is characterized by complex terrain and climate. The SETP
has an average elevation of more than 4000 m above sea level (a.s.l.), and the mean elevation generally
decreases from the inner high-elevation plateau in the northwest to the southern border. According to the
Randolph Glacier Inventory (RGI 6.0) (RGI Consortium, 2017), the SETP hosts many glaciers, with a
total of 7756 glaciers and a total area of 7260 km$^2$. Among these glaciers, 301 glaciers have an area > 4
km$^2$ and account for approximately 50% (3638 km$^2$) of the total glacier area. In the context of global
climate warming, glaciers in the SETP have been shrinking dramatically, which has caused the formation
of numerous glacial lakes in this region (Zhao et al., 2022; Ke et al., 2020). The mean annual temperature
in the SETP generally decreases from southeast to northwest. According to historical meteorological data,
the temperature in the SETP generally ranges from 0 ℃ to 19 ℃, and the hottest and coldest months
are July and January, respectively (Prefecture, 2014). The SETP is affected by the Indian summer
monsoon, the East Asian summer monsoon, and westerlies, which cause the SETP to be one of the most
humid regions across the TP. Precipitation is mostly concentrated from April to September and accounts



for nearly 80% of the annual precipitation in the SETP (Zhao et al., 2022). The seasonality of
precipitation also varies across the region. The southern areas around the Great Bend of Yarlung Tsangpo
are strongly affected by the southeast monsoon, while the high-elevation mountain ranges in the northeast
and northwest are less impacted, resulting in a marked decrease in the mean annual precipitation from
the southwest to the northeast and northwest (Sakai et al., 2015).

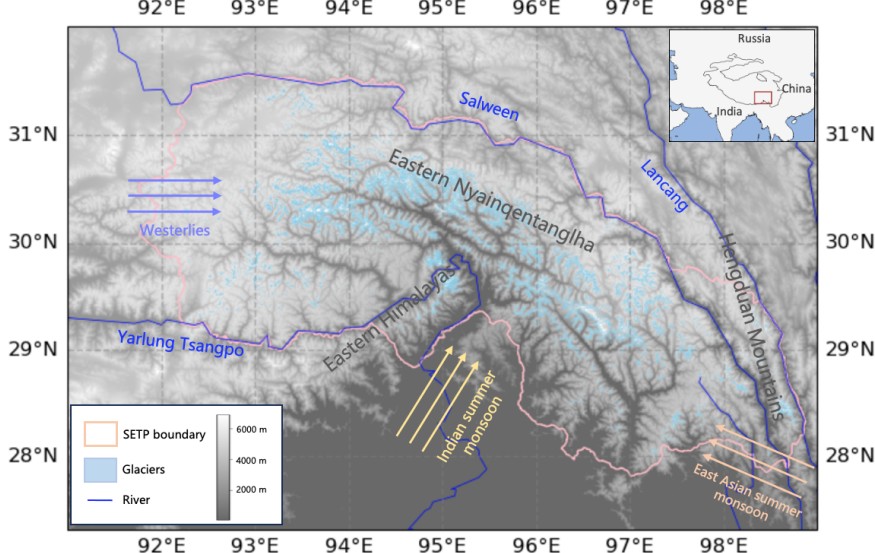

**Figure 1: Overview of the SETP and the distribution of the glaciers.**

**2.2 Dataset**
**2.2.1 ASTER stereo imagery**
The Advanced Spaceborne Thermal Emission and Reflection Radiometer (ASTER; Farr et al., 2007) is
an imaging instrument onboard the Terra satellite that was launched in December 1999. The ASTER
sensor captures high-resolution (15 to 90 m per pixel) images at a temporal resolution of 16 days with
fourteen different bands. Since ASTER can obtain quasireal-time (approximately 55 s difference) stereo
image pairs, it can produce detailed digital terrain models using the digital photogrammetry technique.
In this study, daytime Level-1A (L1A) stereo images (NASA/METI/AIST/Japan Spacesystems and
U.S./Japan ASTER Science Team, 2001) with less than 60% cloud coverage were acquired for time-
series DEM generation within the period 2000-2022.



### 2.2.2 ICESat/ICESat-2 data

The Ice, Cloud, and Land Elevation Satellite (ICESat) is a National Aeronautics and Space Administration (NASA) satellite mission for measuring ice sheet mass balance, cloud and aerosol heights, land topography and vegetation characteristics. ICESat carries the Geoscience Laser Radar Altimeter System (GLAS) and emits infrared and visible laser pulses at 1064- and 532-nm wavelengths with a revisit cycle of 91 days. The Ice, Cloud, and Land Elevation Satellite-2 (ICESat-2) is the second generation of the laser altimeter ICESat mission. ICESat-2 was launched in 2018 and carries the Advanced Topographic Laser Altimeter System (ATLAS) instrument for land elevation measurements. ATLAS emits visible laser pulses at a wavelength of 532 nm, and it generates six beams arranged in three pairs that enable increased coverage of the Earth's surface compared to that of its predecessor, ICESat, which has only one laser beam. In this study, ICESat/GLAH14 data (Zwally et al., 2014) from 2003-2009 and ICESat-2/ALT06 data (Smith et al., 2020) from 2018-2022 were collected through the National Snow & Ice Data Center (NSIDC).

### 2.2.3 CryoTEMPO EOLIS data

The EOLIS (Elevation Over Land Ice from Swath) is part of the European Space Agency (ESA) CryoSat-2 thematic product (CryoTEMPO, Gourmelen et al., 2018), which aims to extend the ability of CryoSat-2 to measure elevation changes in areas of sea ice, polar oceans, land ice, coastal areas, and hydrology. The EOLIS products exploit Swath processing of CryoSat's SARIn mode, which makes full use of the information contained in CryoSat-2's waveforms and provides data with increased spatial and temporal coverage over land ice (Gourmelen et al., 2018). The CryoTEMPO EOLIS consists of two distinct products: a point product containing a cloud of elevations with associated uncertainty in geospatial units and a gridded product containing a spatial interpolation of the point product onto a uniform grid of elevation and uncertainty. The CryoTEMPO-EOLIS point product and gridded product are produced for 14 geographic regions of the Earth. Due to the lack of data available for the SETP region, CryoTEMPO EOLIS point data were collected for our study. Specifically, the CryoTEMPO EOLIS point data provide a monthly elevation dataset with associated uncertainties in $100 \times 100$ km geospatial tiles and temporal coverage from July 2010 until now. We collected the CryoTEMPO-EOLIS point data in the temporal



range from July 2010 to December 2022 in our study, and the data were obtained from https://cryotempo-
eolis.org.

#### 2.2.4 Ancillary datasets

The high-resolution DEMs generated from NASA's Shuttle Radar Topography Mission (SRTM; Farr et
al., 2007) in 2000 were acquired in our study. Specifically, we selected the void-filled version of the
SRTM-GL1 DEM to assist in ASTER DEM generation and time-series elevation change calculations.
To generate the surface water masked DEM image, the Global Surface Water (GSW) dataset derived
from the European Commission's Joint Research Centre (JRC) was collected (Pekel et al., 2016).
Moreover, the GlobeLand30 land cover dataset (Chen et al., 2015) from 2010 was acquired to determine
the stable region of the SETP, and the Randolph Glacier Inventory (RGI 6.0) outlines were used to
identify the glacier region in our study (RGI Consortium, 2017).

#### 3    Methodology

#### 3.1 Time-series DEM generation from ASTER stereo imagery

We used the Ames Stereo Pipeline (Beyer et al., 2018; Shean et al., 2016) tool to create time-series
ASTER DEMs instead of acquiring existing DEM datasets online. The time-series DEM is generated
based on automatic processing with ASTER Level 1A stereo images. Specifically, we processed the
ASTER pairs using the void-filled SRTM-GL1 DEM as a seed DEM for initial orthorectification. The
semiglobal matching (SGM) algorithm (Hirschmuller, 2008) with a kernel size of 9×9 pixels was used
for stereo matching, and the ternary census transform was selected as the cost function (Han et al., 2016).
Subpixel refinement was also implemented to improve stereo matching; specifically, affine window
adaptation with a kernel size of $25 \times 25$ pixels was applied to obtain subpixel disparities. The generated
DEM contains serious errors due to inaccurate stereo matching of the complex landscape or cloud
contamination of the image; then, we removed the outlier elevation pixels with heights that differed by
more than 150 m from the SRTM-GL1 DEM. To generate yearly ASTER DEMs, image mosaicing
processing was carried out for all the generated DEMs of each year to eliminate the holes in the generated
DEM image. To avoid seasonal effects in the time-series analysis of glacier elevation changes, the



generated ASTER DEMs with acquisition dates near July (glacier ablation season) were placed on the
top layer during the ASTER DEM mosaicking. To derive reliable glacier elevation changes based on
multisource elevation datasets, coregistration of the generated ASTER DEMs was carried out. We applied
the method proposed by Nuth and Kääb (2011) to calculate the geometrical shifts, systemic offsets, and
elevation-dependent biases of the generated ASTER DEMs using the SRTM-GL1 DEM as a reference.
The parameter for coregistration was calculated based on the ASTER DEM of the stable region, which
was determined by masking the glacier region, surface water region, forest region, shrub region and
cropland region by using RGI 6.0 data, JRC GSW data and Globeland30 data, respectively.
**3.2 Time-series elevation change calculation**
We performed multisource data processing on the geographical $0.5° \times 0.5°$ tiles of each year, and the
elevation difference between the derived elevation and SRTM-GL1 DEM data was subsequently
calculated for each tile and each year. Altimetry data, including ICESat, ICESat-2 and CryoSat-2 data,
measure surface elevation through spaceborne altimeters, and the derived elevation usually has high
precision (Fricker et al., 2005; Brunt et al., 2019). However, altimeter measurements are sparsely
distributed on the ground surface, which limits the capability of continuous earth surface monitoring.
Therefore, the altimetry data-based elevation difference was calculated for only the locations of altimeter
footprints. The obtained elevation differences had errors caused by either the satellite data measurements
or the SRTM DEM data. To prevent bias in the estimation of elevation change, elevation differences
larger than 150 m were removed from our study. We calculated the mean elevation difference for the
glacier region and stable region on each geographical tile. Since the glacier elevation change varied at
different altitudes, the mean elevation difference was calculated for each 100-m elevation bin of the
glacier region, except for the ICESat data-derived elevation differences, due to the limited amount of
measurements.
**3.3 Estimation of glacier elevation/mass change**
We derived the glacier elevation change in each graphical tile by a glacier area-weighting method based
on the elevation bins of the tile. To remove the systematic error between the derived elevation and SRTM
DEM, we corrected the glacier elevation change based on the elevation change estimate in the stable



region, and the correction was as follows:
$$dh_{glacier-cor} = dh_{glacier} - dh_{stable} \qquad (1)$$
where $dh$ is the elevation difference between the derived elevation and SRTM DEM. We performed
corrections of glacier elevation changes, which were derived from satellite data from different sources
except for the CryoTEMP EOLIS point data since elevation measurements were lacking in the nonglacier
region. To remove the systematic error in elevation change obtained from CryoTEMP EOLIS data, we
adopted an indirect approach; that is, the changes in CryoTEMP elevation were corrected by referring to
the corrected elevation change in the ICESAT-2 data. Specifically, the corrected CryoTEMP elevation
changes are obtained by subtracting the mean difference between the elevation changes derived from the
CryoTEMP and ICESAT-2 data during the overlap period. The correction for the change in CryoTEMP
elevation is as follows:
$$dh_{glacier-cor}^{Cryo} = dh_{glacier}^{Cryo} - \overline{(dh_{glacier}^{Cryo} - dh_{glacier}^{isat2})}_{year} \qquad (2)$$
where $dh_{glacier-cor}^{Cryo}$ and $dh_{glacier}^{isat2}$ are the glacier elevation changes derived from CryoTEMP and
ICESAT-2 data, respectively. $year$ represents the overlapping year of the CryoTEMP and ICESAT-2
measurements. We obtained the corrected elevation change for each geographical tile, and the overall
elevation change for the entire SETP can be obtained with a glacier area-weighted method based on
geographical tiles. Accordingly, the glacier mass change can be calculated through $\Delta M = \Delta V \cdot \rho_g = A \cdot$
$\Delta h \cdot \rho_g$, where $\Delta V$ is the volume of the glacier change, which can be calculated by multiplying the
glacier area $A$ and glacier elevation change $\Delta h$. $\rho_g$ is the glacier density, which was set to
$850\ kg\ m^{-3}$ in our study (Brun et al., 2017).
**3.4 Uncertainty estimation**
The time-series elevation changes for the geographical tiles and the entire SETP region can be obtained
from four satellite images with different temporal coverages. To prevent large biases caused by single
satellite data-derived elevation changes, a simple averaging method is used for fusing multisource data-
derived elevation changes. To estimate the rates of glacier elevation change for each geographical tile
and the entire SETP region, a robust linear fitting using the random sample consensus (RANSAC)



algorithm was carried out for the time-series elevation changes. The uncertainty in the elevation change
rate can be derived from the residuals of the fitted elevation changes, and the calculation is as follows:
$\varepsilon = y_i - y_{fitted}$ (3)
$\sigma_{dh} = \sqrt{\frac{1}{n} \sum_{i=1}^{n} (\varepsilon_i - \bar{\varepsilon})^2}$ (4)
$\sigma_{dh/dt} = \frac{\sigma_{dh}}{dt}$ (5)
where $\varepsilon$ is the residual of the fitted elevation change $y_{fitted}$ and $y_i$ is the original elevation change
corresponding to the $i$-th year. $dt$ and $dh$ are the temporal coverage of the study phase and the overall
elevation change, respectively.
The uncertainty in glacier mass change can be calculated through $\sigma_{\Delta M} = \sqrt{\sigma_{\Delta M,r}^2 + \sigma_{\Delta M,s}^2}$, where $\sigma_{\Delta M,r}$
and $\sigma_{\Delta M,s}$ are the random error and systematic error, respectively. The random errors caused by three
main sources are assumed to be independent: the uncertainty in elevation change $\Delta h$, the uncertainty in
glacier area $\sigma_A$, and the uncertainty in glacier density $\sigma_{\rho_g}$. The uncertainty in the glacier density
$60\ kg\ m^{-3}$ was used in our study (Brun et al., 2017). Then, the random error can be calculated by:
$\sigma_{\Delta M,r} = \sqrt{\left(\sigma_{\Delta v} \cdot \rho_g\right)^2 + \left(\sigma_{\rho_g} \cdot \Delta V\right)^2}$ (6)
where $\sigma_{\Delta V}$ is the uncertainty in the change in glacier volume and can be calculated by:
$\sigma_{\Delta V} = \sqrt{(\sigma_{\Delta h} \cdot A)^2 + (\sigma_A \cdot \Delta h)^2}$ (7)
where $\sigma_A$ is the uncertainty in the glacier area, which we set to $0.1A$ in our study (Brun et al., 2017;
Kääb et al., 2012). To assess the systematic error, we are inspired by Brun et al. (2017) and use the
absolute value of the triangulation residual between two subperiods as the systematic error:
$\sigma_{\Delta M,s} = |\Delta M_{2000-2022} - (\Delta M_{2000-2012} + \Delta M_{2013-2022})|$ (8)
where $\Delta M_{xxxx-yyyy}$ is the glacier mass change for the period between $xxxx$ and $yyyy$.

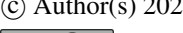


## 4    Results

### 4.1 Glacier elevation changes in the SETP region

We derived glacier elevation changes from multisource satellite data (Fig. 2 (a)). According to the average glacier elevation changes, the glaciers experienced rapid thinning with a total elevation change of $-16.942 \pm 1.06$ m during the period 2000-2022, which corresponded to the glacier mass changes of $-100.70 \pm 13.94 \, Gt$. We divided the period into two subperiods, 2000-2012 and 2012-2022, and the glacier mass changes for the periods 2000-2012 and 2012-2022 were $-46.76 \pm 9.31 \, Gt$ and $-53.20 \pm 8.21 \, Gt$, respectively. The obtained glacier elevation changes varied between satellite data sources. As shown in Table 1, even though all the satellite measurements revealed negative glacial elevation changes for different periods during 2000-2022, the rates of glacier elevation change derived from different satellite data showed significant differences, which demonstrated that estimation errors exist among the single satellite data-derived results. We estimated the rate of glacier elevation change by integrating multisource satellite data. According to Table 1, the obtained rates of glacier elevation change for 2000-2022, 2000-2012, and 2013-2022 were $-0.710 \pm 0.046 \, m/yr$, $-0.583 \pm 0.052 \, m/yr$ and $-0.765 \pm 0.037 \, m/yr$, respectively. The obtained rates of glacier elevation change demonstrated that glacier thinning in the recent decade of 2012-2022 accelerated at a rate of 31.2% compared with that in the previous decade of 2000-2012. The glacier elevation change rate also varied with altitude; as shown in Fig. 2 (b), glacier melting occurred at a faster rate at reduced altitudes. Specifically, the fastest glacier thinning rate was approximately $-1.6 \, m/yr$ at an altitude of approximately 4000 m. When the altitude increased to 4700 m, glacier thinning was maintained at a rate of approximately $-0.5 \, m/yr$ until the altitude reached 5400 m. When the altitude was above 5900 m, the change in glacier elevation became positive.

Table 1. Glacier elevation change rates derived from multisource data.

| Data source | Period | Elevation change rate (m/yr) |
|---|---|---|
| ICESAT/GLAS | 2003-2009 | $-1.179 \pm 0.440$ |
| ICESAT-2/ATL | 2018-2022 | $-0.336 \pm 0.118$ |
| CryoTEMP EOLIS | 2010-2022 | $-1.065 \pm 0.109$ |
| ASTER DEMs | 2000-2022 | $-0.604 \pm 0.296$ |
| **Combined** | **2000-2022** | $\mathbf{-0.710 \pm 0.046}$ |



| | 2000-2012 | $-0.583 \pm 0.052$ |
| | 2012-2022 | $-0.784 \pm 0.035$ |


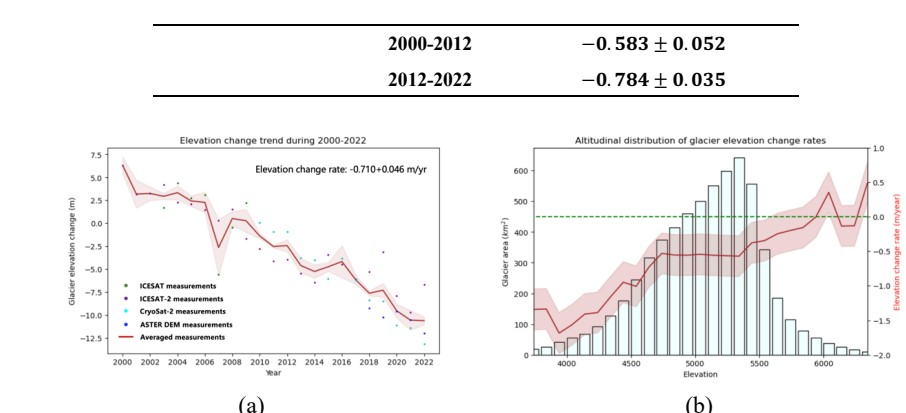

(a)                                    (b)

**Figure 2: Temporal and altitudinal glacier elevation change in SETP. (a) Multisource satellite data-integrated glacier elevation changes during 2000-2022 and (b) the altitudinal distribution of glacier elevation change rates.**

**4.2 Spatiotemporal variability in glacier elevation change**
The total glacier area in the SETP is approximately $7260\ \text{km}^2$, and most of the glaciers are distributed
in the latitudinal range $29°\ N$-$31°\ N$ and the longitudinal range $93°\ E$- $97.5°\ E$ (Fig. 2. (a)). Among
the total $0.5° \times 0.5°$ geographical tiles of the SETP, the tiles of latitude $29°\ N$-$29.5°\ N$ and longitude
$96.5°\ E$-$97°\ E$ cover the largest glacier area of $768\ \text{km}^2$. With a simple statistical analysis of the glacier
area of tiles, 17 tiles had glacier areas larger than $100\ \text{km}^2$, and the aggregated area accounted for 83%
of the total glacier area of the SETP region. Since a smaller glacier area usually corresponds to fewer
satellite measurements and thus introduces great uncertainty in glacier elevation change calculations, we
selected 17 geographical tiles with glacier areas larger than $100\ \text{km}^2$ for quantitative analysis in our
study. According to Fig. 3 (a) and (c), the fastest glacier thinning events occurred at latitude $29°\ N$-
$29.5°\ N$ and longitude $97°\ E$-$97.5°\ E$, and the slightest glacier thinning occurred at latitudes $30°\ N$-
$30.5°\ N$ and longitudes $93.5°\ E$ - $94°\ E$, corresponding to glacier thinning rates of $-0.930 \pm$
$0.164\ \text{m/yr}$ and $-0.421 \pm 0.211\ \text{m/yr}$, respectively. We further estimated the change in glacier
elevation between 2000-2012 and 2012-2022. The glacier elevation change rates and their difference
values for the two periods are illustrated in Fig. 3. (c) and (b), where 12 geographical tiles showed larger
negative changes from 2012-2022 than from 2000-2012; in general, these findings revealed that the
glaciers thinned at an accelerating rate in approximately 70% of the region in the SETP in the past decade.



Specifically, the glacier thinning rates were less than 1 m/yr for all the geographical tiles from 2000–
2012; however, in the next decade from 2012–2022, 9 geographical tiles experienced glacier thinning
rates higher than 1 m/yr, and the fastest glacier thinning rate reached 2.151 m/yr, which demonstrated
a serious accelerated glacier thinning rate in the SETP region.

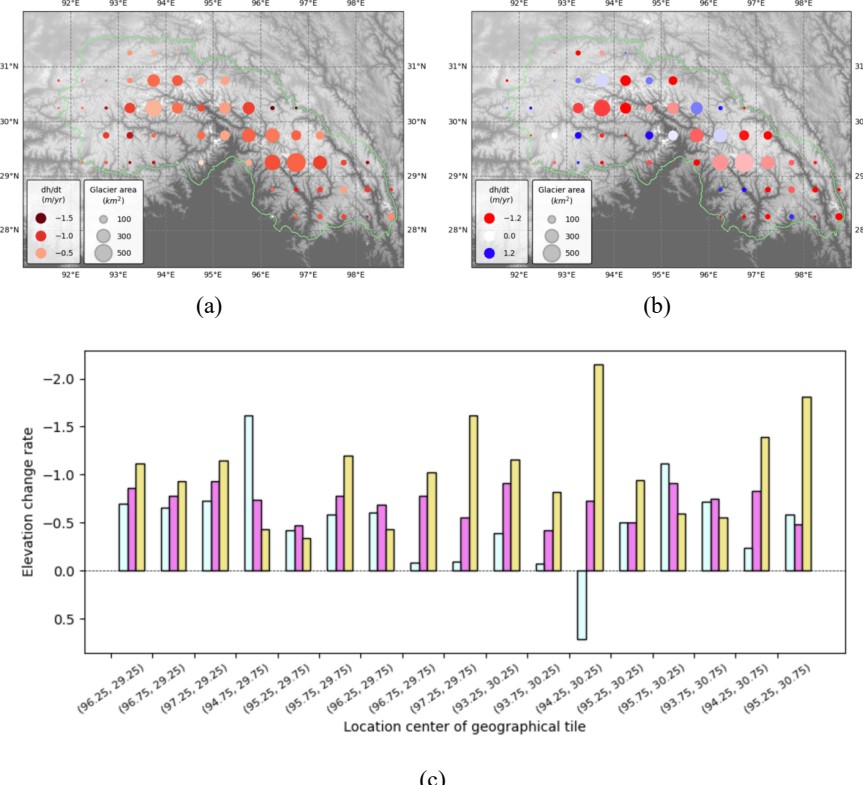

**Figure 3: Spatial distribution of the glacier elevation change rate. (a) The spatial distribution of glacier elevation change rate during 2000-2022, and (b) the spatial distribution of the difference of rates of glacier elevation change between 2000-2012 and 2012-2022.**

**4.3 Individual glacier changes**
We estimated the rate of elevation change for individual glaciers and explored the variability in the rate
of change in glacier elevation in the SETP. A total of 301 glaciers with areas larger than $4\ km^2$, which
accounted for 51% of the glacier area in the SETP, were selected for analysis in our study. As shown in
Fig. 4, most of the glaciers demonstrated negative elevation changes, while a small portion of the glaciers
still exhibited a positive elevation change during the period 2000-2022. The glacier thinning rate varies



with spatial distribution and glacier area; to explore the features of elevation change in individual glaciers,
we selected individual glaciers corresponding to different elevation change ranges for further analysis.
Specifically, we divided the glaciers into three types, namely, regions with glacier accumulation,
moderate glacier thinning rate, and rapid glacier thinning rate, which corresponded to elevation change
rates larger than 0 m/yr, between -1 m/yr and -0 m/yr, and smaller than -1 m/yr, respectively. The glacier
area proportions of the three types of glaciers accounted for 12.6%, 56.5%, and 30.9%, respectively.
Specifically, the glaciers of RGI60-15.12221 and RGI60-15.11833 were selected for glacier
accumulation analysis. According to Fig. 4 (b), these two glaciers experienced positive elevation changes
in the context of glacier thinning trends during the period 2000-2022 in the SETP. In particular, the
RGI60-15.11833 glacier achieved a high glacier accumulation rate of 0.81 m/yr during the period of
2000-2022. According to the visualization of the two glacier accumulation cases shown in Fig. 4 (b),
both glacier thinning and glacier accumulation occurred, and glacier accumulation was dominant in the
glaciers. We selected the glaciers RGI60-15.01098 and RGI60-15.01098 for the analysis of moderate
glacier thinning rates. These two glaciers exhibited similar general trends of glacier elevation change,
and the rates of glacier elevation change were -0.67 m/yr and -0.72 m/yr, respectively. The elevation
change maps shown in Fig. 4 (c) illustrate that negative elevation changes were dominant in the
individual glaciers. In addition to having a moderate glacier thinning rate, some glaciers experienced
rapid negative elevation changes during the period of 2000-2022. We selected two glaciers, RGI60-
15.11957 and RGI60-15.12221, to represent rapid glacier thinning. The rates of glacier elevation change
for the two selected glaciers reached to -1.13 m/yr and -1.68 m/yr, and most regions of the individual
glaciers experienced severe negative elevation changes, as shown in Fig. 4 (d).





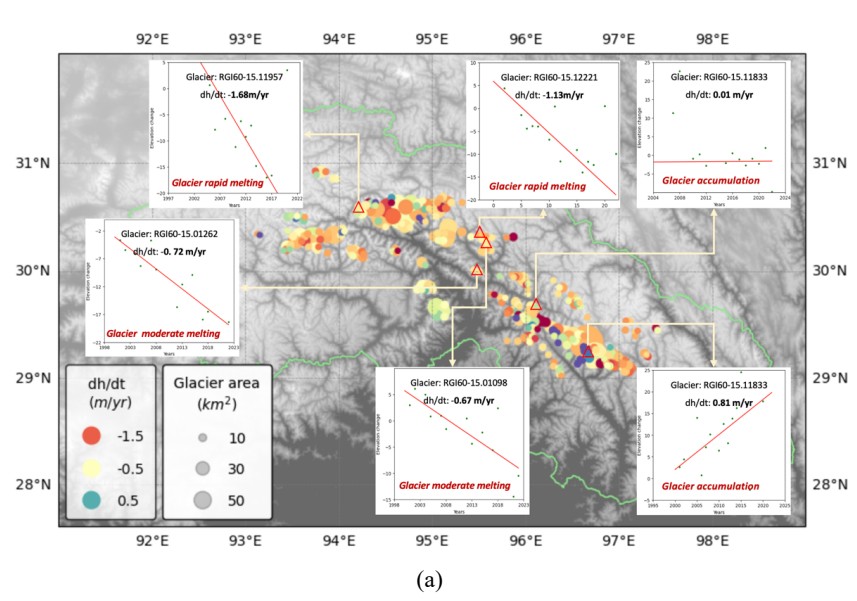

(a)

(b) Accumulated glacier

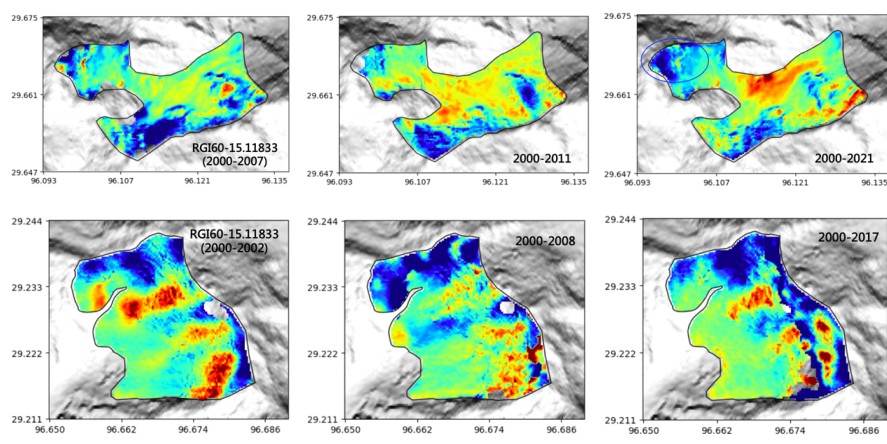

(c) Moderate melted glacier

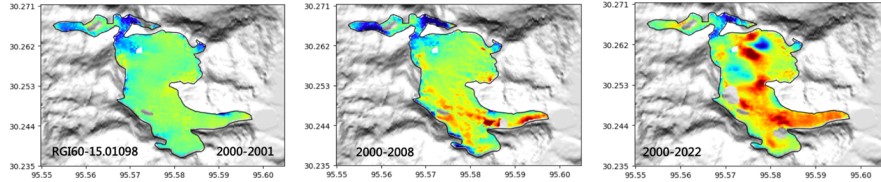



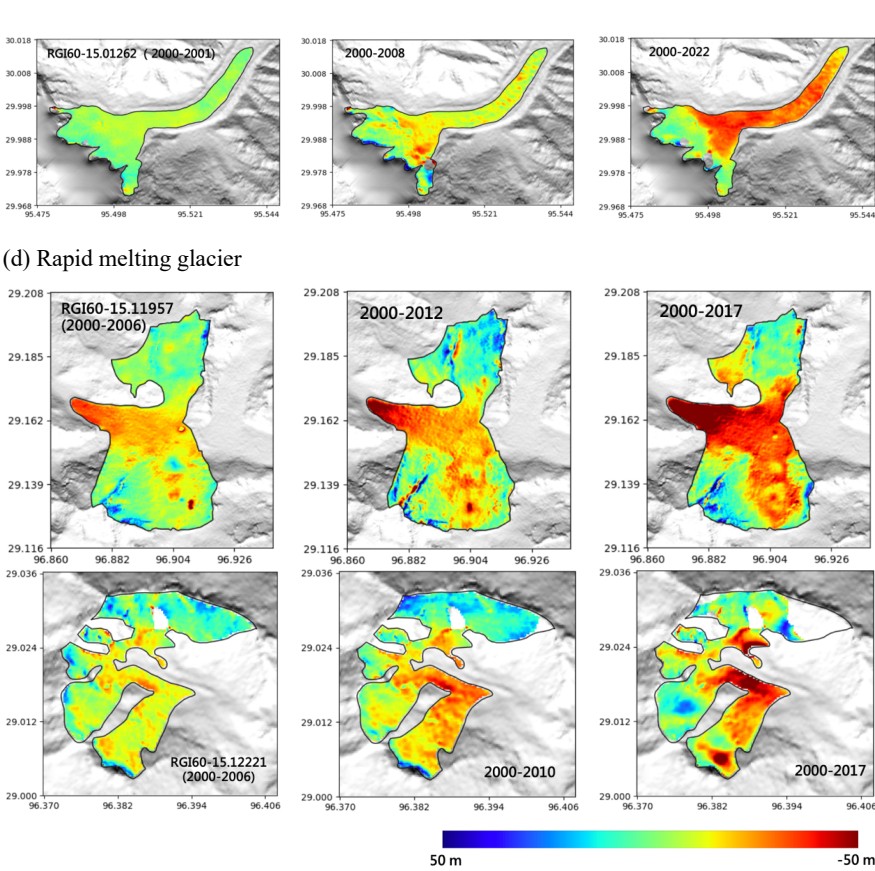

**Figure 4: Spatial distribution and elevation change rates of individual glaciers. (a) The spatial distribution of elevation change rates for individual glaciers. (b), (c), and (d) correspond to the glacier accumulation, moderate glacier melting rate, and rapid glacier melting rate cases, respectively.**

## 5    Discussion
### 5.1 Cross-analysis of multisource satellite measurements
We evaluated the quality of the elevation measurements through a cross-analysis of the results derived
from different satellite sources. We assumed that the error of the DEM was spatially consistent and
random; then, the correlation between elevation measurements derived from different satellites could be
analyzed with the probability distributions of elevation difference values between the elevation
measurements and the base DEM. Specifically, we calculated the elevation difference values using either
the SRTM DEM or ASTER DEM as the base DEM and for both the stable and glacier regions. According



to the probability distribution curves shown in Fig. 5 (a), the elevation measurements of the ASTER
DEM were strongly correlated with the ICESat data-derived elevation measurements in the stable region,
and both showed slight positive biases compared with those of the SRTM DEM data. We analyzed the
correlation between different satellite measurements in the glacier region, as shown in Fig. 5 (c). The
elevation measurements between ICESat and CryoSat-2 were strongly correlated, and the elevation
measurements of ICESat, ICESat-2 and CryoSat-2 were greater than those of the ASTER DEM. We
found that the elevation measurements of ICESat-2 were the smallest both in the stable region and in the
glacier region. We linked this situation to the observation scale of the satellite sensors, meaning that the
spatial resolution of the derived ASTER DEM was 30 m, and the footprint diameters of ICESat, ICESat-
2 and CryoSat-2 were 70 m, 17.5 m, and 300 m, respectively. Among these satellites, ICESat-2 had the
finest spatial observation scale. We compared the elevation measurements of the stable region with those
of the glacier region. According to Fig. 5 (b) and (c), the precisions of the elevation measurements
obtained by satellite sensors were different between the stable region and glacier region, indicating that
the elevation measurements from the ASTER DEM were more consistent with the altimetry data in the
stable region; this difference may be mainly due to the relatively flat terrain in the stable region.

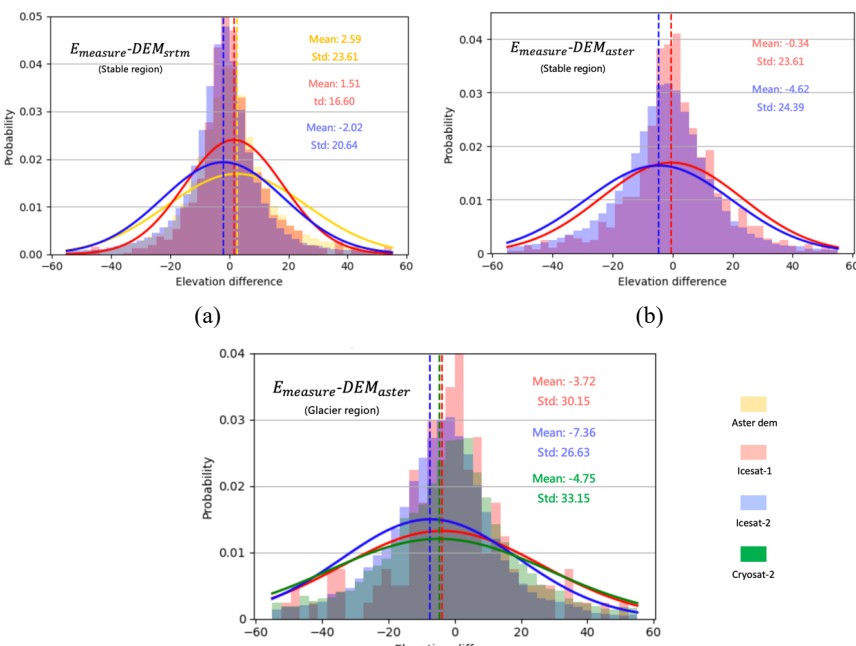

(c)

**Figure 5: Cross analysis of multisource satellite measurements. The histograms are probability distributions of elevation difference values derived by satellite measurement and the DEM image. The colors of histograms are corresponding to different satellite data sources. In addition, the elevation difference values are obtained from the stable region and the glacier region, respectively.**

The ICESat-2 satellite carries ATLAS, which features new technologies that enable it to collect more
detailed and precise elevation measurements of the Earth's surface. The ICESat-2 collects the
measurements by emitting pairwise beams, and each beam pair consists of a strong and a weak beam.
For the CryoSat-2 swath altimetry data, which were also mentioned as CryoTEMPO-EOLIS point data
in our study, swath processing fully exploits the information contained in CryoSat-2 waveforms and
leads to one to two orders of magnitude more measurements than the CryoSat-2 L2 data processed by
the point-of-closest-approach (POCA) technique. Accordingly, we evaluated the performance of the
strong beam and weak beam measurements of the ICESat-2 data, as well as the CryoSat-2 swath and L2
measurements of the CryoSat-2 data in this study. We calculated the elevation difference between the
ICESat-2 data and the ASTER DEM in the stable region, as shown in Fig. 6 (a). The ICESat-2 strong
beam and weak beam had no significant differences in elevation measurements. CryoSat-2 swath and L2
measurements were taken on the glacier region. According to Fig. 6 (b), the CryoSat-2 swath
measurements were relatively consistent with the ASTER DEM measurements, while the CryoSat-2 L2
measurements showed large differences from the ASTER DEM measurements. Accordingly, we can
conclude that the CryoSat-2 swath data achieve significant improvement over the CryoSat-2 L2 data in
terms of glacier elevation measurements.

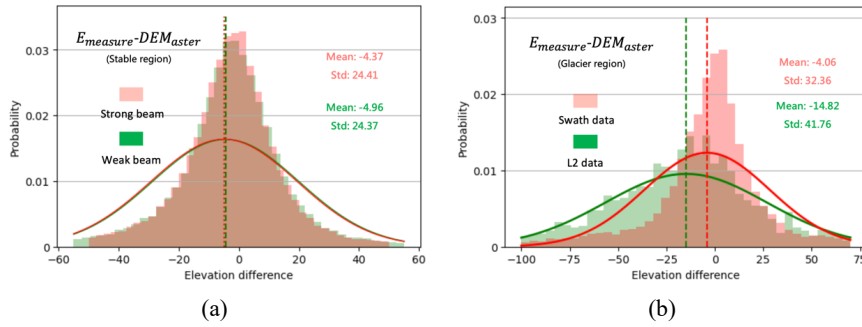

(a)                                    (b)

**Figure 6: Comparisons of homologous elevation measurements. The elevation difference is calculated using the elevation measurements and ASTER DEM image. (a) Probability distribution histograms of elevation**





**differences for the ICESat-2 strong beam measurements and weak beam measurements. (b) Probability distribution histograms of elevation differences for the CryoSat-2 swath measurements and CryoSat-2 L2 measurements.**

**5.2 Comparison of existing estimates of glacier elevation change**
Many researchers have exploited the rate of glacier elevation change in the SETP region, which is similar
to the region of the Nyainqentanglha range or the Hengduan Mountains (Gardner et al., 2013; Neckel et
al., 2014; Yi et al., 2020; Shean et al., 2020). According to the estimates shown in Fig. 7 (a), studies
show large disparities among the estimated glacier elevation change rates, and the largest and smallest
negative elevation changes correspond to -1.34 m/yr and -0.4 m/yr, respectively (Kääb et al., 2015;
Gardner et al., 2013). Among the 16 collected estimates of glacier elevation change rates, 12 estimates
ranged from -0.9 m/yr to -0.5 m/yr. Our estimates also fell within this scope and were consistent with the
recent research by Hugonnet et al. (2021); specifically, the rates of glacier elevation change for 2000-
2012, 2012-2022, and 2000-2022 were $-0.58 \pm 0.05$ m/yr, $-0.78 \pm 0.04$ m/yr, and $-0.71 \pm 0.05$
m/yr, respectively, which were similar to the estimates of $-0.55 \pm 0.28$ m/yr, $-0.81 \pm 0.41$ m/yr,
and $-0.69 \pm 0.35$ m/yr for the periods 2000-2009, 2010-2019, and 2000-2019, respectively, by
Hugonnet et al. (2021). Our estimate for the period of 2000-2022 was also very similar to the estimates
of Zhao et al. (2022) and Brun et al. (2017), who demonstrated that glaciers melted at rates of $-0.71 \pm$
0.18 m/yr and $-0.73 \pm 0.27$ m/yr during the periods of 2000-2019 and 2000-2016, respectively.
To accurately capture accurate glacier elevation changes, different satellite data sources have been
explored for quantifying glacier elevation changes in recent studies. The ICESat data have been widely
used for estimating the rate of glacier elevation change for the period of 2003-2009, as shown in Fig. 7.
(b) The rates of glacier elevation change derived from ICESat data have great disparities (Gardner et al.,
2013; Neckel et al., 2014; Kääb et al., 2015). The CryoSat-2 interferometric-swath-processed data, which
were recently applied for estimation of the rate of glacier elevation change, are estimated for the period
of 2011-2019 to be $-1.15 \pm 0.12$ m/yr (Jakob et al., 2021), which seems to indicate a slightly more
rapid rate of glacier thinning than other studies. The estimates obtained from time-series ASTER DEMs
seem to fall within a reasonable range between -0.55 m/yr and -0.81 m/yr (Hugonnet et al., 2021 and
Brun et al., 2017). Generally, among the single data source-based estimates of glacier elevation change

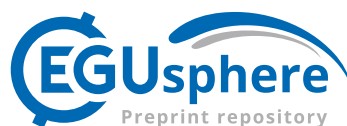

rates, the ASTER DEM-derived results exhibited high consistency, while the ICESat data-derived results
varied greatly. We believe that this difference is mainly due to the spatially continuous observation of
the ASTER stereo images, which provide more abundant elevation measurements than the dispersed
observation of the ICESat data and thus result in more robust estimates. With the increase in accessible
satellite data, additional satellite data have been integrated to improve the estimation of glacier elevation
change rates. Multisource data integration can enhance glacier observations at both spatial and temporal
scales and reduce bias caused by anomalous measurements from single-source data. We integrated the
ICESat, ICESat-2, and ASTER DEMs and CryoSat-2 data and achieved improved estimates of the rate
of glacier elevation change with wider temporal coverage and lower uncertainty. More accessible satellite
data can be combined for the estimation of glacier elevation changes, and we believe that a high-
efficiency combination strategy and improved satellite data processing technique will be crucial factors
for future improved estimations.

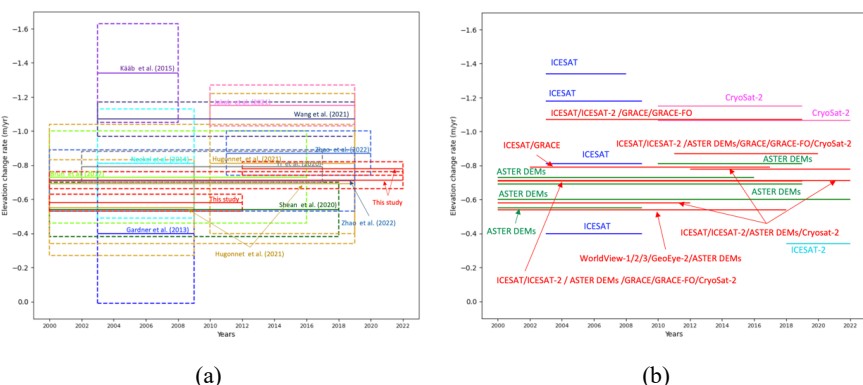

(a)                                        (b)

**Figure 7: Estimates of glacier elevation change rate in published studies. (a) The glacier elevation change rates with estimate uncertainties and (b) the data source for the estimation of glacier elevation change rate.**

**6. Conclusion**
We integrate multisource satellite data (including ASTER DEM, ICESat GLAH14, ICESat-2 ATL06,
and CryoSat-2 CryoTEMPO-EOLIS point data) to estimate the glacier elevation changes across the
entire SETP region over the past two decades. We found that glaciers are experiencing rapid and
heterogeneous elevation thinning in the SETP, and the mean glacier thinning rate is $-0.710 \pm$



0.046 $m/yr$. In particular, the glacier thinning rates for the periods 2000-2012 and 2012-2022 were
estimated, and we found that the glacier thinning rate in the recent decade of 2012-2022 accelerated by
31.2% compared with that in the previous period of 2000-2012.
The estimates for individual glaciers show large disparities; that is, the proportions of accumulating
glaciers, moderate melting glaciers and rapid melting glaciers account for 12.6%, 56.5%, and 30.9%,
respectively. We compared the newly accessible CryoSat-2 swath data with the CryoSat-2 L2 data and
found that the elevation measurements from the CryoSat-2 swath data agreed significantly more with the
elevation measurements from other satellite observations. According to extensive cross-analyses of
glacier elevation measurements derived from different satellite sources, we found that the glacier
elevation measurements derived from the new advanced ICESat-2 satellite were slightly smaller than
those derived from other satellite observations. Among the existing studies on the estimates of glacier
elevation changes in the SETP region, our estimates are highly consistent with recent studies and have
higher temporal resolution and lower estimate uncertainty.
**Code/Data availability**
The ICESat GLAH and ICESat-2 ATL06 used in this study are available from the National Snow and
Ice Data Center (NSIDC). The CryoSat-2 CryoTEMPO-EOLIS point data were obtained from
https://cryotempo-eolis.org.    The    aster    stereo    images    were    acquired    at
https://search.earthdata.nasa.gov/search. The datasets generated in this study are available from the
authors upon request. The processing code is available at https://github.com/xinluo2018/Glacier-in-
SETP.
**Author contributions**
Xin Luo developed the methods and wrote the paper. Hongping Zeng and Zhen Ye assisted with the
data processing.
**Competing interests**
The authors declare that they have no conflicts of interest.



**Acknowledgments**

This work is supported by the National Natural Science Foundation of China (Proj No. 42001351).
Sentinel-1 SAR images were obtained from the European Space Agency.

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
