# Peer review of "Investigating the spatiotemporal features of glacier"

_EGUsphere, 2023_

## Author Comment (AC1)

**Response to Referee #2:**

Dear Referee:

Many thanks for your review of our paper. The paper has been carefully revised according to your comments, and detailed point-by-point responses to the comments are given below. In our letter, the comments from the reviewer are highlighted in bold font and our responses listed below are in black font.

**General Comments**

*1)The authors investigated the spatiotemporal features of glacier elevation changes over the southeastern Tibetan Plateau using multisource satellite data, it is an interesting work.*

R.  Thank you for your comment. The point-by-point responses to your specific comments are given below.

*2)While the current work is not sufficient. The authors introduced that glacier in the SETP experienced accelerated melting in recent decades in the Introduction. However, based on multisource satellite date investigation and comparison with previous studies, they got the same result, and then the research is over. There have no novel findings, even though they did a lot of work.*

R. Thank you for your comments. The previous studies and our study both demonstrate that the SETP are experiencing accelerated melting in recent decades, however, the obtained glacier elevation changes and the estimation uncertainties are varying. In our study, the multi-source satellite data including the new released data such as ICESat-2 ALT06 data and CryoSat-2 CryoTEMPO EOLIS data, are integrated for the glacier elevation measurement, which significantly improve the estimation of glacier elevation change by compared with the previous study.

For the contributions of our study, 1) we introduced a new method for yearly glacier elevation change estimation by integrating ASTER stereo image, ICESat,

ICESat-2, and CryoSat-2 CryoTEMP-EOLIS altimetry data, and our estimation result achieved finer spatiotemporal resolution and smaller uncertainty compared with the existing estimation results. Due to the effective integration of the multisource satellite data in our study, the improved estimation result for the glacier elevation change has been obtained. We think the new proposed method also provide a valuable reference for the future glacier study in the other region of the earth. 2) We carried out extensive analysis for the multisource satellite measurements. In particular, we compared the ICESat-2 measurements by the strong beam with the weak beam, and we revealed that the ICESat-2 strong beam and weak beam had no significant differences in glacier elevation measurements. We also compare the CryoSat-2 CryoTEMP-EOLIS data with the commonly used CryoSat-2 L2 data in our study, and we found the CryoSat-2 CryoTEMP-EOLIS data achieves significant improvement over the CryoSat-2 L2 data in glacier elevation measurement. 3) We derived the glacier elevation change for the overall SETP region at a fine spatiotemporal resolution. Accordingly, we analyzed the spatiotemporal variability of the glacier elevation change in SETP region, and we found the SETP region not only contains the melting glaciers but also contains the accumulated glaciers. 4) We further analyze the anomalous glaciers (accumulating glaciers) in term of glacier altitude, glacier aspect, glacier number, glacier slope and glacier length, respectively. Generally, the glacier accumulation occurs in 38 glaciers that account for 12.62% of the selected 301 glaciers. We compared the characteristics of the accumulating glaciers with the melting glaciers in our study. We found that most of the accumulating glaciers are facing the southwestern side and the melting glaciers are mainly facing the northern and eastern sides (North, Northeast, East, Southeast), and the accumulating glaciers characterized in general by steeper glacier slope, shorter glacier length, and slight lager altitude compared with the melting glaciers.

More details have been added to the Section 4.2 of the revised manuscript. We think these findings could further improve our knowledge and understanding for the glacier elevation change in the SETP region. The related description has been added to Section 4.2 and Section 5.1 of the revised manuscript.

*SPECIFIC COMMENTS*

*1) In addition, there have many confusions in the Introduction. For example, in the second paragraph, the authors introduced that glaciers in the HMA are greatly retreating, and then list several research and numbers, there have no summarize. And then the authors said that glacier elevation changes in the SETP have not been well quantified by existing studies due to inconsistent results and give some examples. I know the authors want to emphasize the importance of glacier elevation change in the SETP and it should have drawn more attention. However, more research and numbers were given to emphasize the inconsistence of previous studies in the end of this paragraph. It is a confusion to me.*

R. Thank you for your comments. To make a clearer expression for the second paragraph of the Introduction section, we have reorganized this paragraph.  Specifically, 1) we firstly introduce the rapid glacier melting trend in HMA region, then 2) we illustrate that the glacier elevation changes show large spatial variability in HMA based on specific studies on different subregions. 3) We mentioned that among the subregions the glaciers in the SETP region have experienced the strongest recession in HMA since 2000. Finally, 4) we demonstrate that the current estimates for the glacier elevation changes in SETP region is still limited in accuracy, and then show the inconsistent results derived from the previous studies. The revised paragraph has been added to the Introduction section in our revised manuscript.

"

1 Introduction

…

The High Mountain Asia (HMA) region hosts the largest glacier concentration outside the polar regions, and these glaciers prominently contribute to streamflow in one of the most populated areas of the world (Brun et al., 2017). Current studies have revealed that the glaciers in HMA are greatly retreating (Brun et al., 2017; Gardner et al., 2013; Yao et al., 2012). For example, Brun et al. (2017) illustrated that the glacier elevation change rate for the total HMA region was -0.21 ± 0.05 m/yr during 2000-2016, and

Shean et al. (2020) revealed that the total HMA glacier mass change rate during 2000-2018 was −0.19 ± 0.03 m w.e./yr. Glacier elevation change shows large spatial variability in HMA. Specifically, a positive glacier elevation change rate of 0.047 ± 0.06 m/yr was reported in Kunlun by Shean et al. (2020), a slight negative glacier elevation change rate of -0.1± 0.06 m/yr was observed in Karakoram by Kääb et al. (2015), and the most negative elevation change rate of -0.73 ± 0.27 m/yr was found in Nyainqentanglha by Brun et al. (2017). Among the subregions the glaciers in the SETP region have experienced the strongest recession in HMA since the 2000s, and this recession in the SETP accounts for approximately one-quarter to one-third of the total glacier mass loss in HMA (Brun et al., 2017; Kääb et al., 2015; Neckel et al., 2014). Glacier elevation changes in the SETP have drawn the most attention from glaciologists, climatologists, and geoscientists, and great efforts have been made to improve elevation change estimations (Brun et al., 2017; Gardner et al., 2013; Shean et al., 2020). However, estimates of glacier elevation changes are still limited in accuracy and are inconsistent with each other for the SETP region. For example, different estimates of − 0.40 ± 0.41 m/yr for 2003–2009 (Gardner et al., 2013), − 0.81 ± 0.32 m/yr for 2003–2008 (Neckel et al., 2014), − 0.68 ± 0.22 m/yr for 2000–2016 (Brun et al., 2017), − 1.11 ± 0.11 m/yr for 2010–2019 (Jakob et al., 2021), − 0.54 ± 0.16 m/yr for 2000–2018 (Shean et al., 2020), − 0.69 ± 0.35 m/yr for 2000–2019 (Hugonnet et al., 2021), and − 0.73 ± 0.18 m/yr for 2003–2020 (Zhao et al., 2022) were reported in existing studies. "

*2) In the third paragraph, the authors summarized the approaches for glacier observation through spaceborne remote sensing. The first approach is satellite images, including glacier outlines delineation from Landsat, Sentinel, and glacier elevation determination from satellite-based optical stereo images. The second approach is geodetic digital elevation model differencing, including InSAR and optical stereo images. That is a repeat.*

R. Thank you for your comments. We have rephrased the literatures review about the glacier observation approaches in the third paragraph. Specifically, we modified the first type 'satellite image' to 'Nadir optical and SAR satellite images', thus the optical

stereo image-based method is removed from this type. The related modification has been added to the revised manuscript.

"

1 Introduction

…

Glacier observations are mainly obtained through spaceborne remote sensing. There are four major approaches for observing glaciers: 1) Nadir optical and SAR Satellite images: These images have long been used for glacier observation and have provided abundant historical glacier change information along with climate change information (Ke et al., 2015). Mountain glaciers can typically be automatically or semiautomatically delineated with satellite images, such as Landsat, Advanced Spaceborne Thermal Emission and Reflection Radiometer (ASTER), Sentinel-2, or Sentinel-1 images (Pope and Rees, 2014, Robson et al., 2020; Zhou and Zhen, 2017; Peng et al., 2023). 2) Geodetic digital elevation model (DEM) differencing: DEM differencing is the most common approach for calculating the elevation change in an entire glacier area and enables the estimation of glacier mass change over time (Cogley et al., 2011; Robson et al., 2022). Generally, spaceborne optical and radar sensors are two main sources for producing DEMs for calculating elevation changes in large-scale glacierized regions. Spaceborne radar DEMs, such as those from the Shuttle Rader Topography Mission (SRTM) C-band data and TanDEM X-band data (Ke et al., 2020; Guan et al., 2022), are generated using interferometric synthetic aperture radar (InSAR). The radar signal penetrates snow and ice, and the penetration depth is a matter of debate and leads to uncertainty in glacier elevation estimation (Barundun et al., 2015; Kääb et al., 2015). Spaceborne optical digital models (DEMs) are generated using stereo images (Bhushan et al., 2021; King et al., 2023;); the generated DEMs are free of ice/snow penetration, while optical stereo images are unavailable in cloudy regions.

"

***3) About the data source, the specific time of DEM acquired from ASTER images is not clear, and Fig 2(a) showed the glacier elevation changes from ICESat, ICESat2, CryoSat and ASTER DEM, the ICESat2 measurements (the purple dots) covered from 2002 to 2020, how it could be?***

R. Thank you for your comments. The ICEsat-2 data acquired from 2018 to 2022 and the ASTER data acquired from 2000-2022 are used in our study. It is a mistake for the legend color in our previous manuscript, specifically, the ICESat-2 measurements should be blue dots, and the ASTER DEM measurements should purple dots. The Fig 2 (a) have been revised in our revised manuscript as follows.

[Figure]

Figure 2 (a): The spatial distribution of glacier elevation change rate during 2000-2022

To leverage ASTER images as much as possible, all the ASTER stereo images with cloud coverage less than 60% are acquired for ASTER DEMs generation in our study. And in total 1671 pair-wise ASTER stereo images are eventually acquired for the time-series DEM generation in our study. To avoid seasonal effects in the analysis of annual glacier elevation changes, we make the priority of the generated ASTER DEMs which are acquired near July (glacier ablation season) for the glacier elevation measurement of the year. Specifically, the specific times of DEM acquired from ASTER images corresponding to different years are shown below.

[Figure]

Figure.1: Specific time of DEMs derived by ASTER stereo images. The subtitle of each figure is the correspondng year of the generated DEM.

*4) Therefore, the potential of this work is great, but considerable work is required before making it publishable.*

R. Thank you for your comments. we have carefully revised our paper according to your constructive comments and suggestions. Generally, the Introduction section particularly for the literature review about the glacier elevation change in the HMA region and SETP region and the related methods has been thoroughly revised. In the

Section of Datasets used, the more detailed information about the used satellite data has been added to the revised manuscript. In the Methodology section, we have further described the detailed method about the multisource satellite data integration in glacier elevation change estimation. In the Results section, we changed the two periods of 2000-2012 and 2012-2022 to 2000-2011 and 2011-2022 to make the two periods the same years. In addition to illustrate that some anomalous accumulating glaciers appeared in the SETP region, we have further analyzed the accumulating glaciers features in terms of glacier altitude, glacier aspect, glacier number, glacier slope and glacier length, respectively. In the Discussion section, the bias among the satellite measurements and the comparison with the existing estimates have been presented with improved expression. In the last section of Conclusion, we conclude the works and related results of our study, and we also described the limitation of our study and our future study plan. Accordingly, we believe that the revised version of manuscript has been significantly improved.

Once again, thank you for your time reviewing this paper.

Best wishes,

The authors

---

## Author Comment (AC2)

**Response to Referee #1:**

Dear Referee:

Many thanks for your review of our paper. The paper has been carefully revised according to your constructive comments, and detailed point-by-point responses to the comments are given below. In our letter, the comments from the reviewer are highlighted in bold font and our responses listed below are in black font.

*General Comments*

*1)The paper deals with an important topic which is the estimation of mass balance of mountain glaciers by a combination of different satellite-based methods. It is focused on the SE of the Tibetan Plateau which is a region with fast glacier down-wasting rates. This region was part of several studies focusing on the whole of the High Asia, Tibetan Plateau, or Himalaya-Karakoram region using typically just a single remote sensing method. This laborious study brings some interesting results in terms of the trends of glaciers in the studied region and the differences among the used metods. However, there are several serious shortcomings in the presented study.*

R. Thank you for your comment. We have carefully revised our paper according to your comments and suggestions, and a point-by-point reply to your comment is given below.

*2)The study does not provide necessary details on the application of the widely used methods which are applied differently by different authors. The authors also failed to show the reasons behind the differences amongst the methods.*

R. Thank you for your comment. We divide the methods for glacier monitoring into four categories, that is, 1) Nadir optical and SAR satellite image; 2) Geodetic digital elevation model (DEM) differencing; 3) Satellite altimetry; and 4) Satellite gravity. The Geodetic DEM differencing and satellite altimetry are the widely used methods for the glacier elevation change monitoring.

The detailed method for deriving glacier elevation change usually varies with the

specific application and the used satellite data. Specifically, Kääb et al. (2015) investigated the glacier thickness changes in the SETP region by using ICESat satellite altimetry data for 2003-2008. They derived glacier elevation difference trends at a $1° \times 1°$ geographic grid by fitting a robust linear temporal trend to the time series of elevation differences between the SRTM DEM and individual ICESat footprint elevation. Hugonnet et al. (2021) used an improved techniques of MicMac for ASTER DEMs generation, and then apply a multi-step outlier filtering to iteratively improve the quality of DEMs. The filtering algorithm consists of a spatial filter, removing elevation outside a topographical maximum and minimum from the TanDEM-X elevations in the pixel surroundings, and a temporal filter propagated from the TanDEM-X elevation at a given pixel through a maximum possible glacier elevation change rate. Shean et al. (2020) produced the high-resolution DEMs based on stereo imagery acquired by multiple satellite platforms, such as WorldView-1/2/3, GeoEye-1, and ASTER, between 2000 and 2018. The acquired satellite stereo images were processed with the NASA Ames Stereo Pipeline (ASP) (Shean et al., 2016; Beyer et al., 2018) using the void-filled SRTM-GL1 product as a seed DEM for initial orthorectification. And the semi-global matching (SGM) correlator (Hirschmuller, 2008) with default parameters ($7 \times 7$ pixel window) was adopted to improve the results for scenes with limited image resolution and/or texture. Zhao et al. (2022) integrated satellite altimetry (ICESat, CryoSat-2, and ICESat-2) with differencing and satellite gravity to resolve high spatiotemporally resolved glacier elevation change across SETP during 2000-2020. The authors generated DEMs from ASTER L1A stereo-images to estimate the multi-year elevation change, which was used to correct for the altimetry-based counterpart. Moreover, the authors processed three different satellite altimetry data and total water storage (TWS) products, and finally, the time series of glacier elevation change for period of 2000–2020 was obtained through the integration of satellite altimetry-, satellite gravity-, and DEM-derived elevation changes.

For the possible reasons behind the differences amongst the methods, we think it is mainly due to the different data sources used. Generally, the basic techniques for glacier elevation measurement varies with different satellite data, for example, the

measurement by ASTER stereo images is based on the digital photogrammetry, and the measurement by ICESat data is based on the full-waveform laser altimetry, accordingly, the corresponding data processing methods are varying.

The related description has been added to the Introduction section of the revised manuscript.

**Reference:**

Kääb, A., Treichler, D., Nuth, C., Berthier, E.: Contending estimates of 2003–2008 glacier mass balance over the Pamir–Karakoram–Himalaya, The Cryosphere, 9, 557–564, 2015.

Hugonnet, R., McNabb, R., Berthier, E.: Accelerated global glacier mass loss in the early twenty-first century, Nat., 592 (7856): 726-731, 2021.

Shean, D.E., Bhushan, S., Montesano, P.: A systematic, regional assessment of high mountain Asia glacier mass balance. Front. Earth Sc-switz., 2020, 7: 363, 2020.

Beyer, Ross, A., Oleg, A., Scott, M.: The Ames Stereo Pipeline: NASA's open source software for deriving and processing terrain data, E.S.S., 5, 2018.

Hirschmuller, H. (2008). Stereo processing by semiglobal matching and mutual information. IEEE Trans. Pattern Anal. Mach. Intell. 30, 328–341.

Zhao, F., Long, D., Li, X.: Rapid glacier mass loss in the Southeastern Tibetan Plateau since the year 2000 from satellite observations, Remote Sens. Environ., 270: 112853, 2022.

*3) The study does not refer to any ground measurements that could support the superiority of one of the approaches and validate the overall results on specific glaciers. There is a paper by Yao et al. (2012) that includes ground measurements of glacier mass balance on the Tibetan Plateau from which two glaciers should fall into the extent of the presented study. Additionally, newer measurements could be available meanwhile.*

*Reference*
*Yao, T, Thompson, L., Yang, W., Yu, W., Gao, Y., Guo, X., ... & Joswiak, D. (2012). Different glacier status with atmospheric circulations in Tibetan Plateau and surroundings. Nat Clim Change 2: 663–667.*

R. Thank you for your comment. We have carefully read the paper by Yao et al. (2012), specifically, two glaciers of Parlung No.12 Glacier ($29° \, 18'N, 96° \, 54'E$) and

the Ata Glacier ($29° 10' N, 96° 48' E$) mentioned in this paper are fall into the extent of our study region. The paper illustrates that the Ata Glacier experienced a fast glacial length reduction exceeded 80 m yr$^{-1}$ in 2005/2006, however, the in-situ observation for glacier elevation change is not provided. The field observations about the glacier thickness have been carried out for the Parlung No.12 Glacier for the period of 2006-2010 based on ground-penetrating radar measurements, however, this glacier cannot be found in our Randolph Glacier Inventory (RGI 6.0) (RGI Consortium, 2017) data. As reported by the authors, the area of this glacier is only 0.21 km$^2$ in 2005 and the authors predict that this glacier will disappear very soon due to the fast glacier retreating. Beyond this paper mentioned the two field measurements in our study region, we didn't find another ground measurements which could be used for the result validation in our study.

The time-series glacier elevation changes are derived by integrating multisource satellite data in our study. To validate the estimation of time-series glacier elevation changes, the time-series ground measurements are required. However, the glacier regions are usually located in remote mountainous areas which is difficult for the human to reach. Accordingly, to validate the estimation of glacier elevation changes, a cross-validation method is usually adopted. For example, Kääb et al. (2015) validated their ICESat-derived glacier elevation changes by using the satellite gravimetry-derived result. Zhao et al. (2022) validated their multisource satellite observations-derived results by using the results of the previous studies. In our study, we also use a cross-validation method, that is, the glacier elevation measurement by ASTER DEM and satellite altimetry is evaluated by using the SRTM DEM data, and the overall time-series glacier elevation changes are evaluated by using the results derived from the previous studies.

The mentioned references have been added to our revised paper, and the related description are highlighted in the revised manuscript.

**Reference:**

Yao, T., Thompson, L., Yang, W., Yu, W., Gao, Y., Guo, X., ... & Joswiak, D. (2012). Different glacier status with atmospheric circulations in Tibetan Plateau and surroundings. Nat Clim Change 2: 663–667.

RGI Consortium, 2017. Randolph Glacier Inventory - A Dataset of Global Glacier Outlines, Version 6. [Indicate subset used]. Boulder, Colorado USA. NSIDC: National Snow and Ice Data Center.

Jacob, T., Wahr, J., Pfeffer, W. T., and Swenson, S.: Recent contributions of glaciers and ice caps to sea level rise, Nature, 482, 514–518, 2012.

Kääb, A., Treichler, D., Nuth, C., Berthier, E.: Contending estimates of 2003–2008 glacier mass balance over the Pamir–Karakoram–Himalaya, The Cryosphere, 9, 557–564, 2015.

Zhao, F., Long, D., Li, X.: Rapid glacier mass loss in the Southeastern Tibetan Plateau since the year 2000 from satellite observations, Remote Sens. Environ., 270: 112853, 2022.

*4) The quality of the study would largely improve if the authors would focus on the reasons behind the differences, that the particular methods produce and on the glaciers with positive mass balance. An analysis of these anomalous glaciers in terms of accumulation area altitude, orientation of the glaciers, and feeding mechanisms would be highly beneficial. Such analysis would provide enough material to allowing the discussion to get more interesting. The section Conclusions in this form is rather weak.*

R. Thank you for your comment. We investigated the glacier elevation changes over the SETP region based on multisource satellite data, which contains ICESat, ICESat-2, CryoSat-2, and ASTER stereo image. Accordingly, the time-series glacier elevation changes with high spatiotemporal resolution are obtained. With the obtained results, the in-depth analysis for the spatiotemporal features of glacier elevation changes over the SETP region was carried out, generally, we revealed that most of the glaciers show negative elevation change, while there are remain a small portion of glaciers exhibited an anomalous positive elevation change during the period of 2000-2022.

According to your suggestion, we further analyze the anomalous glaciers (accumulating glaciers) in term of glacier altitude, glacier aspect, glacier number, glacier slope and glacier length, respectively. Generally, as shown in Fig.4 (b), the glacier accumulation occurs in 38 glaciers that account for 12.62% of the selected 301

glaciers. We compare the features of the accumulating glaciers with the melting glaciers in our study. As shown in Fig.4 (a), most of the melting glaciers are facing the northern and eastern sides (North, Northeast, East, Southeast) while most of accumulating glaciers are facing the southwestern side. According to Fig.4 (c), (d) and (e), the accumulating glaciers characterized in general by steeper glacier slope, shorter glacier length, and slight lager altitude by comparing with the melting glaciers.

[Figure]

Figure 4: Comparative analysis of the accumulating glacier and melting glacier in terms of (a) glacier aspect, (b) glaciers number, (c) glacier slope, (d) glacier length, and (e) glacier altitude in the SETP region. The N, NE, E, SE, S, SW, W, and NW represent North, Northeast, East, Southeast, South, Southwest, West and Northwest, respectively.

It is very meaningful to reveal that the feeding mechanisms of the glacier accumulation. The SETP region locates at the intersection of the westerlies, the Indian Ocean monsoon, and the southeast monsoon, so that this region has abundant precipitation and is the most humid area across the TP. Therefore, we infer the main feeding of the glacier accumulation is the precipitation such as the snowfall in this region. It also raises the question that why only a small portion of glaciers occur glacier accumulation and the most of glaciers are experiencing melting in the SETP. We suppose that it is due to the complex mountain-valley terrain which divides the SETP region into local zones with great different climates, the glacier melting occurs in the

most zones that are affected by increased temperature with the global warming trend, and glacier accumulation occurs in some specific zones which remain keep low temperatures. In this study, we revealed the appearance of anomalous glacier accumulation and the corresponding features of the accumulated glaciers in the SETP region, and the feeding mechanisms of the glacier accumulation is also analyzed. Our conclusion still needs to be further validated with continuous ground observation of meteorological data in both the glacier accumulation zones and the glacier melting zones, since it will take a long time for the collection of ground data, therefore we plan the meaningful study in our further work.

The related revision has been added to the revised manuscript and is shown below.

"

4.2 Spatiotemporal variability in glacier elevation change

…

We further analyze the anomalous glaciers (accumulating glaciers) in term of glacier altitude, glacier aspect, glacier number, glacier slope and glacier length, respectively. Generally, the glacier accumulation occurs in 38 glaciers that account for 12.62% of the selected 301 glaciers as shown in Fig.4 (b). We compared the characteristics of the accumulating glaciers with the melting glaciers in our study. As shown in Fig.4 (a), most accumulating glaciers are facing the southwestern side and the melting glaciers are mainly facing the northern and eastern sides (North, Northeast, East, Southeast). According to Fig.4 (c), (d) and (e), the accumulating glaciers characterized in general by a steeper glacier slope, short glacier length, and a slight lager altitude by comparing with the melting glaciers. Since the SETP region locates at the intersection of the westerlies, the Indian Ocean monsoon, and the southeast monsoon, this region has abundant precipitation and is the most humid area across the TP. Therefore, we infer the main feeding of the glacier accumulation is the precipitation such as the snowfall in this region. The most of glaciers are experiencing melting in the SETP region, it is

mainly due to the complex mountain-valley terrain which divides the SETP region into local zones with great different climates, and the glacier accumulation appears in specific zones which remain keep low temperatures, and the glacier melting occurs in the most zones that are affected by increased temperature under the global warming trends.

…

"

**SPECIFIC COMMENTS**

**1) 19 'recent decade' to 'last decade compared to the previous one.**

R. Thank you for your suggestion. We have modified the previous decade and the last decade to the periods of 2000-2011 and 2011-2022, respectively, thus making the expression more explicit. The related modification has been added to the revised manuscript.

"

ABSTRACT

…

We divided the study period into two decades of 2000-2011 and 2011-2022, and found that glacier thinning accelerated at a rate of 31.4% in the period of 2011-2022 by comparing with the period of 2000-2011.

"

**2) 36 'downstream humans' to 'downstream population'**

R. Thank you for your suggestion. The 'downstream humans' has been modified to 'downstream population' in our revised paper.

*3) 58 remove 'the most'*

R. Thank you for your comment. According to your suggestion, 'the most' has been removed from our revised manuscript.

*4) 68 Add 'Optical and SAR' before 'satellite images'*

R. Thank you for your comment. According to your suggestion, 'Optical and SAR' has been added to our revised manuscript.

*5) 69 'along the climate change': Do the satellite images really contain direct information about climate change? Rephrase the sentence.*

R. Thank you for your comment. The satellite images provide abundant historical glacier change information, and we think the glacier change is mainly caused by the climate change, therefore, we said the satellite images provide information along with the climate change. According to your suggestion, we have removed the 'along the climate change', and the related description has been added to the revised manuscript as follows.

"

1. Introduction

…

These images have long been used for glacier observation and have provided abundant historical glacier change information (Ke et al., 2015).

"

*6) 80 'large scale' to 'large'*

R. Thank you for your comment. According to your suggestion, the 'large scale' has been replaced to 'large' in our revised manuscript.

*7) 82 'using interferometric synthetic aperture radar': better to write 'using SAR interferometry'*

R. Thank you for your comment. According to your suggestion, the 'using interferometric synthetic aperture' has been replaced to 'using SAR interferometry' in our revised manuscript.

*8) 83 'matter of debate': Better write that the penetration can reach meters and that it depends mainly on the presence of liquid water in the superimposed snowpack and ice.*

R. Thank you for your comment. According to your suggestion, the related description has been added to our revised manuscript.

"

Introduction

…

The radar signal penetrates snow and ice, and the penetration can reach meters which depends mainly on the presence of liquid water in the superimposed snowpack and ice, and thus leads to uncertainty in glacier elevation estimation (Barundun et al., 2015; Kääb et al., 2015).

"

*9) 87 Optical satellite stereo processing is also limited in areas of fresh snow which lacks features for feature matching.*

R. Thank you for your comment. According to your suggestion, the related description has been added to the Introduction section of the revised manuscript.

"

1 Introduction

…the generated DEMs are free of ice/snow penetration, while optical stereo images are limited in areas of fresh snow which lacks features for feature matching and are unavailable in cloudy regions.

"

**10) 87b (ICESat missions): There was only one ICESat mission. The instrument was however switched on for several time periods (campaigns) due to the lack of energy.**

R. Thank you for your comment. According to your suggestion, the 'ICESat missions' has been modified to the 'ICESat mission' in our revised manuscript.

**11) 91-93 Specify whether you mean lidar or radar altimetry here.**

R. Thank you for your comment. The satellite altimetry here includes both LiDAR and Radar altimetry. The related description has been added to the revised manuscript.

"

1 Introduction

… Satellite altimetry includes both LiDAR and Radar altimetry has been an efficient approach for investigating glacier elevation changes due to the continuous temporal observation of the Earth's surface;

"

**12) 99 Remove 'Above all' as it does not match here.**

R. Thank you for your comment. According to your suggestion, the 'Above all' has been removed in the revised manuscript.

**13) 111-115 This you can list as objectives**

R. Thank you for your comment. According to your suggestion, the related sentences have been listed as objectives of our study, and the related description has been added to the revised manuscript.

"

1 Introduction

… The main objectives of this study contain as follows: 1) To improve the estimation of glacier elevation change by fully exploiting the ASTER stereo image, ICESat, ICESat-2 and CryoSat-2 altimetry data. 2) To investigate the spatiotemporal features of glacier elevation changes in the SETP and to quantify the heterogeneity of glacier elevation changes in this region. 3) To perform comprehensive cross-analysis for elevation measurements taken by multisource satellites and estimates of glacier elevation changes obtained from existing studies.
"

*14) 116-120 You do not need this. Clear chapter headings and sub-headings are enough to get quickly a good overview of the study.*

R. Thank you for your comment. According to your suggestion, we have removed the related sentences in our revised manuscript.

*15) 129 This means that the mean glacier area is about 1km. Is this true?*

R. Thank you for your comment. It is true that the mean glacier area is about 1km$^2$, specifically, there are 6482 glaciers with the area smaller than 1 km$^2$, which accounts for 83.57% of the total glaciers. On the other hand, the area summation of the 6482 glaciers is 1811.01 km$^2$, which only accounts for 24.94% of the total glaciers area. The histogram statistic of glacier areas is shown below, and the related description has been added to the revised manuscript.

[Figure]

Figure 1: Histogram statistic of glacier areas.

*16) 134: Better to provide information on the mean annual temperature and its changes throughout the studied area. Some figures on total precipitation and basic facts about its regional variation would also be useful.*

R. Thank you for your comment. We acquired the meteorological data at a spatial resolution of 0.25° latitude by 0.25° longitude for the SETP region during the period of 2000-2021. According to the data analysis, we found that the mean temperature of the SETP region is increasing from about 1℃ to 3℃ during 2000-2021 (Wu, 2017). The precipitation also varies across the region, that is, the mean annual precipitation decreases from the southeast to the northwest and northeast, and the yearly precipitation varies from about 500 mm to 1300 mm (Wu et al., 2013). According to your suggestion, the figures about the annual temperatures and the spatially resolved precipitation map are provided in the revised manuscript as follows.

[Figure]

Figure A: Temperature and precipitation of the SETP region. (a) Annual averaged temperatures of the SETP region during 2000-2021. (b) Spatially resolved averaged precipitation map during 2000-2021.

"

**2.1 Study area**

…

The mean temperature of the SETP region is increasing from about 1℃ to 3℃ during 2000-2021 (Wu, 2017). The SETP is affected by the Indian summer monsoon, the East Asian summer monsoon, and westerlies, which cause the SETP to be one of the most humid regions across the TP. Precipitation is mostly concentrated from April to September and accounts for nearly 80% of the annual precipitation in the SETP (Zhao et al., 2022). The precipitation also varies across the region, that is, the mean annual precipitation decreases from the southeast to the northwest and northeast, and the yearly precipitation varies from about 500 mm to 1300 mm (Wu et al., 2013).

…

[Figure]

Figure 1: Overview of the SETP and the distribution of the glaciers. The insets (a) and (b) are of respective the spatially resolved precipitation map and annual temperature by averaging the data acquired from 2000-2021.

"

**Reference:**

Wu J, Gao X, Giorgi F, et al. Changes of effective temperature and cold/hot days in late decades over China based on a high resolution gridded observation dataset[J]. International Journal of Climatology, 2017, 37: 788-800.
Wu J, Gao X. A gridded daily observation dataset over China region and comparison with the other datasets[J]. Chinese Journal of Geophysics, 2013, 56(4): 1102-1111.

*17) How about the accumulation type of the glaciers? See Maussion et al. 2014 for details on this.*

*Reference*
*Maussion, F., Scherer, D., Mölg, T., Collier, E., Curio, J., & Finkelnburg, R. (2014). Precipitation seasonality and variability over the Tibetan Plateau as resolved by the High Asia Reanalysis. Journal of Climate, 27(5), 1910-1927.*

R. Thank you for your comment. According to the reference paper, the study SETP region is a large region of mixed type glaciers, specifically, the MAM/DJF type glacier occupies the largest proportion of the total glaciers, which is followed by the JJA type glacier, and the MAM/JJA type glacier also occupies a small proportion in this region. The related description has been added to the revised manuscript.

"

2.1 Study area

…

According to the glaciers classification based on precipitation seasonality (Maussion et al., 2014), the MAM/DJF (March-May/December-February) type glacier occupies the largest proportion of the total glaciers, which is followed by the JJA (June-August) type glacier, and the MAM/JJA (March-May/June-August) type glacier also occupies a small proportion in this region.

"

**18) 140 Does the SE Monsoon really reach the great bend of Yarlung Tsangpo? Can you support this with a citation?**

R. Thank you for your comment. This viewpoint is come from two papers by Sakai et al. (2015) and Ke et al. (2020), respectively. After carefully reading the reference papers, we also think this viewpoint is not solid enough, therefore, we decide to remove this sentence in our revised manuscript.

**Reference:**

Sakai, A., Nuimura, T., Fujita, K., Takenaka, S., Nagai, H., Lamsal, D., Climate regime of Asian glaciers revealed by GAMDAM glacier inventory, The Cryosphere, 9, 865–880, 2015.

Ke L, Song C, Yong B, et al. Which heterogeneous glacier melting patterns can be robustly observed from space? A multi-scale assessment in southeastern Tibetan Plateau[J]. Remote sensing of environment, 2020, 242: 111777.

"

2.1 Study area

…

The precipitation also varies across the region, that is, the mean annual precipitation decreases from the southeast to the northwest and northeast, and the yearly precipitation varies from about 500 mm to 1300 mm (Wu et al., 2013).

…

"

**19) 143 better to call this section 'Datasets used'**

R. Thank you for your comment. According to your suggestion, the 'Dataset' has been replaced to 'Datasets used'.

**20) 148 'fourteen different bands': better provide information on the bands that acquire the stereo pairs.**

R. Thank you for your comment. According to your suggestion, we have provided more information about the bands in the revised manuscript as follows.

"

2.2.1. ASTER stereo imagery

…

Specifically, the ASTER image consists of four visible and near infrared (VNIR) bands with a resolution of 15 m, six shortwave infrared bands with a resolution of 30 m, and five thermal infrared bands with a resolution of 90 m.

…
"

**21) 148b 'You probably mean 'almost simultaneously' by the 'quasireal-time'. You can remove it as it's not needed here.**

R. Thank you for your comment. According to your suggestion, the 'quasireal-time' has been removed from our revised manuscript. The related description is shown below.
"

2.2.1 ASTER stereo imagery

…

Since ASTER can obtain stereo image pairs and thus produce detailed digital terrain models using the digital photogrammetry technique.

…

"

**22) 151 The limit of 60% of cloud cover is rather high. Would you get too few stereo pairs if you selected a lower threshold such as 30%? Please provide details on this.**

R. Thank you for your comment. It required at least about 82 ASTER images to full cover the SETP region. We set the cloud cover threshold to 30% and searched the ASTER image for each year during 2000-2020, we found that many areas of the SETP are not covered by the acquired ASTER image. In fact, although to set a lager threshold of 60%, there are still data missing in many areas of the SETP region. As shown in Fig.1 below, we select two years of 2005 and 2011 as example, for the two years which have acquired 112 and 52 images by setting the cloud coverage threshold of 60%, while

still couldn't cover the full SETP region. On the other hand, if the cloud cover to be set larger, the ASTER image will be affected by high percentage cloud cover and result in poor image quality. Accordingly, we finally determine the threshold of 60% in our study. The number of ASTER images corresponding to different cloud coverage thresholds are shown in Table 1 and Table 2. The related description has been added to the revised manuscript.

Table 1 ASTER images of different cloud coverage thresholds, for 2000-2012

| threshold/year | 2000 | 2001 | 2002 | 2003 | 2004 | 2005 | 2006 | 2007 | 2008 | 2009 | 2010 | 2011 | 2012 |
|---|---|---|---|---|---|---|---|---|---|---|---|---|---|
| <30% | 19 | 41 | 38 | 53 | 39 | 62 | 35 | 66 | 35 | 32 | 52 | 43 | 49 |
| <60% | 38 | 64 | 65 | 71 | 55 | 112 | 62 | 89 | 76 | 42 | 73 | 52 | 81 |

Table 2 ASTER images of different cloud coverage thresholds, for 2013-2022

| threshold/year | 2013 | 2014 | 2015 | 2016 | 2017 | 2018 | 2019 | 2020 | 2021 | 2022 |
|---|---|---|---|---|---|---|---|---|---|---|
| <30% | 44 | 56 | 66 | 42 | 54 | 44 | 39 | 68 | 53 | 38 |
| <60% | 61 | 82 | 117 | 58 | 98 | 81 | 74 | 92 | 74 | 56 |

2005                    2011

Figure B. The distribution of the derived ASTER measurements by setting the cloud coverage threshold of 60%.

**23) 158 The campaigns are more important for ICESat than the revisit time.**

R. Thank you for your comment. According to your suggestion, the information about ICESat campaigns have been added to the revised manuscript. The related description is shown below.

"

2.2.2 ICESat/ICESat-2 data

… The ICESat operated in an 8-day repeat orbit from 20 February to 4 October 2003. Subsequently, the orbit transitioned to an approximately 30-day operation periods (campaigns), three times per year. During the 7-year operational period of ICESat, extensive data were collected during 20 campaigns, mostly during February/March, May/June, and October/November (Zhang et al., 2019).

…

**References:**

Zhang G, Chen W, Xie H. Tibetan Plateau's lake level and volume changes from NASA's ICESat/ICESat-2 and Landsat Missions [J]. Geophysical Research Letters, 2019, 46(22): 13107-13118.

"

*24) 164 'collected through' to 'obtained from'*

R. Thank you for your comment. According to your suggestion, the 'collected through' has been replaced to 'obtained from' in the revised manuscript.

*25) 166 You should better introduce the dataset in terms of satellite, period of operation etc.*

R. Thank you for your comment. According to your suggestion, more information about the CryoTEMPO EOLIS data has been added to the revised manuscript. The related description is shown below.

"

2.2.3 CryoTEMPO EOLIS data

Under the European Space Agency (ESA) Earth Observation program, Cryosat-2 is loaded with Ku band SAR/Interferometric Radar Altimeter (SIRAL). CryoSat-2 orbits

the Earth with a 369 day near-repeat period formed by the successive shift in a 30-day sub-cycle. The satellite has an inclination of 92°, offering improved coverage of the polar regions. The EOLIS (Elevation Over Land Ice from Swath) is part of the ESA CryoSat-2 thematic product (CryoTEMPO, Gourmelen et al., 2018), which aims to extend the ability of CryoSat-2 to measure elevation changes in areas of sea ice, polar oceans, land ice, coastal areas, and hydrology.

…

"

*26) 176 'lack of data available': Not clear which data are not available.*

R. Thank you for your comment. We mentioned the CryoTEMPO-EOLIS point product and gridded product here. Specifically, for the SETP region the CryoTEMPO-EOLIS gridded product is not available.

*27) 184 'acquired' to 'used'*

R. Thank you for your suggestion. The 'acquired' has been replaced by 'used' in the revised manuscript.

*28) 188 'was acquired' to 'was used'*

R. Thank you for your suggestion. The 'was acquired' has been replaced by 'was used' in the revised manuscript.

*29) 189 'stable region' to 'stable off-glacier region'*

R. Thank you for your suggestion. The 'stable region' has been replaced by 'stable off-glacier region' in the revised manuscript.

*30) 212 It is not clear which landcover class is regarded as a 'stable region' out of the listed classes.*

R. Thank you for your comment. We determine the stable off-glacier region by removing the non-stable region and glacier region in our study. Specifically, the nonstable regions are thought to include surface water region, forest region and shrub region, which are removed by using the obtained JRC GSW data and Globeland30 data. And the glacier region is removed by using the obtained RGI 6.0 data. Accordingly, the remaining region is determined as the stable off-glacier region in our study. The related description has been added to the revised manuscript.

"

3.1 Time-series DEM generation from ASTER stereo imagery

…

The parameter for coregistration was calculated based on the ASTER DEM of the stable off-glacier region, which was determined by masking the glacier region, surface water region, forest region, shrub region and cropland region by using RGI 6.0 data, JRC GSW data and Globeland30 data, respectively.

"

*31) 221 'altimetry data-based' sounds strange. Write better 'elevation differences based on altimetry data*

R. Thank you for your suggestion. The 'altimetry data-based elevation differences' has been modified to 'elevation differences based on altimetry data' in the revised manuscript.

*32) 221b what do you mean by 'location'? The footprint, measurement point, area? Specify.*

R. Thank you for your comment. The 'location' here means the footprint of altimeter, and the related modification has been added to the revised manuscript.

*33) 223 What bias do you want to prevent? Noise, clouds, fog?*

R. Thank you for your comment. Since the elevation change is estimated by using SRTM DEM data and satellite altimetry data, therefore, either the error of SRTM DEM

measurement or the error of satellite altimetry measurement will cause estimation bias. Specifically, the radar altimetry is capable of operating through clouds and its signal is not affected by the clouds and fog, the LiDAR altimetry data used in our study is the high-level processed product, which has been processed with the removal of invalid data, like the cloud contaminated data. Therefore, we think the error of satellite altimetry measurement is mainly caused by the steep terrain of the study region and the high-level data processing method. The SRTM DEM is generated by using Synthetic Aperture Radar (SAR) data, the error of SRTM DEM is still not caused by the clouds and fog and we think it is mainly caused by the steep terrain and the occurrence of voids (Kolecka and Kozak, et al., 2014). The related description has been added to the revised manuscript.

**References:**

Kolecka N, Kozak J. Assessment of the accuracy of SRTM C-and X-Band high mountain elevation data: A case study of the Polish Tatra Mountains[J]. Pure and Applied Geophysics, 2014, 171: 897-912.

*34) 226 'elevation range' instead of 'elevation bin'*

R. Thank you for your comment. According to your suggestion, the 'elevation bin' has been replaced by 'elevation range' in the revised manuscript.

*35) 238-240 Rephrase, not clear.*

R. Thank you for your comment. According to your suggestion, we have rephrased the related sentences in the revised manuscript.

"

3.3 Estimation of glacier elevation/mass change

…

The systematic error led to a global offset for the estimated multi-year glacier elevation

changes, therefore, we directly corrected the multi-year glacier elevation changes derived by CryoTEMP EOLIS data by using the multi-year glacier elevation changes derived by ICESat-2. Specifically, the multi-year glacier elevation changes derived by CryoTEMP data are corrected through subtracting the global offset between the multi-year elevation changes derived by the CryoTEMP and ICESat-2 data in the overlap period.

"

***36) 244 The formula is not clear. Is the line above the difference an average? If yes of what? The variables are named in an awkward way, simplify it.***

R. Thank you for your comment. The line above the difference is an average in our study, and the averaged value represents the global offset between the multi-year elevation changes derived by the CryoTEMP and ICESat-2 data, respectively. According to your suggestion, the variables name have been modified in the revised manuscript. The related description for the formula is shown below.

"

3.3 Estimation of glacier elevation/mass change

…

The correction for the multi-year elevation changes derived by CryoTEMP is as follows:

$$dh_{cryo-cor} = dh_{cryo} - \frac{\sum_{years}(dh_{cryo} - dh_{isat2})}{years} \qquad (2)$$

where $dh_{cryo}$ and $dh_{isat2}$ are the glacier elevation changes derived from CryoTEMP and ICESat-2 data, respectively. $dh_{cryo-cor}$ is the corrected elevation change derived by CryoTEMP data, $years$ represents the overlapping years between the CryoTEMP and ICESat-2 periods.

"

***37) 248 the formula should be definitely on a separate line and numbered.***

R. Thank you for your comment. According to your suggestion, the formula has been placed on a separate line and we assigned a new number to this formula. The related description has been added to the revised manuscript.

**38) 252 What reason is behind the selection of this ice density value? (Other values are also in use.)**

R. Thank you for your comment. We set the ice density value of 850 kg m$^{-3}$ by following most of the related studies. According to literature review, the researches by Brun et al. (2017), Kääb et al. (2015), Hugonnet et al. (2021), Shean et al. (2020), Huss (2013) and Ke et al. (2020) all set the ice density value of 850 kg m$^{-3}$ except for the study by Zhao et al. (2022) which set the ice density value of 900 kg m$^{-3}$. The ice density value of 850 kg m$^{-3}$ obviously have bias with the truth value, therefore, we set an ice density uncertainty of 60 kg m$^{-3}$ during the glacier mass estimation. The related description has been added to the Section 3.3 and Section 3.4 of the revised manuscript.

**Reference:**

Brun, F., Berthier, E., Wagnon, P.: A spatially resolved estimate of High Mountain Asia glacier mass balances from 2000 to 2016, Nat. Geosci., 10 (9): 668-673, 2017.

Zhao, F., Long, D., Li, X.: Rapid glacier mass loss in the Southeastern Tibetan Plateau since the year 2000 from satellite observations, Remote Sens. Environ., 270: 112853, 2022.

Kääb, A., Treichler, D., Nuth, C., Berthier, E.: Contending estimates of 2003–2008 glacier mass balance over the Pamir–Karakoram–Himalaya, The Cryosphere, 9, 557–564, 2015.

Hugonnet, R., McNabb, R., Berthier, E.: Accelerated global glacier mass loss in the early twenty-first century, Nat., 592 (7856): 726-731, 2021.

Shean, D.E., Bhushan, S., Montesano, P.: A systematic, regional assessment of high mountain Asia glacier mass balance. Front. Earth Sc-switz., 2020, 7: 363, 2020.

Huss, 2013. Density assumptions for converting geodetic glacier volume change to mass change. Cryosphere Discuss. 7, 219–244.

Ke, L., Song, C., Yong, B.: Which heterogeneous glacier melting patterns can be robustly observed from space? A multi-scale assessment in southeastern Tibetan Plateau, Remote Sens. Environ., 242: 111777, 2020.

**39) 255 Specify, what bias you mean.**

R. Thank you for your comment. We mean the estimation bias of elevation change here. We have rephrased the related sentences in the revised manuscript as follows.

"

3.4 Uncertainty estimation

…

To prevent the estimation of elevation change to be affected by measurement bias of a single satellite, we used a simple averaging method for fusing multisource data-derived elevation changes.

"

**40) 285 Why not use the time periods 2000-2011 and 2011-2022 to make them the same length? If not, use the yearly difference which is comparable in any case.**

R. Thank you for your comment. According to your suggestion, we have modified the time periods to 2000-2011 and 2011-2022, thus make the two periods the same length. The rates of glacier elevation change for periods of 2000-2011 and 2011-2022 are shown in Table 3. And the yearly glacier elevation differences as well as the linearly fitted elevation differences by using the SRTM data are given in Table 4 below. Additionally, all the results derived by our study including the glacier elevation differences for each 0.5° × 0.5° geographical tile will be open accessed through the website of our open-source project on https://github.com/xinluo2018/Glacier-in-SETP. All the related revision has been added to the revised manuscript.

Table 3. Glacier elevation change rates derived from multisource data.

| Data source | Period | Elevation change rate (m/yr) |
| --- | --- | --- |
| ICESAT/GLAS | 2003-2009 | $-1.179 \pm 0.440$ |
| ICESAT-2/ATL | 2018-2022 | $-0.336 \pm 0.118$ |
| CryoTEMP EOLIS | 2010-2022 | $-1.065 \pm 0.109$ |
| ASTER DEMs | 2000-2022 | $-0.604 \pm 0.296$ |
| **Combined** | **2000-2022** | $\mathbf{-0.710 \pm 0.046}$ |
| | **2000-2011** | $\mathbf{-0.574 \pm 0.054}$ |

| | 2011-2022 | $-0.754 \pm 0.034$ |
|---|---|---|

Table 4. Glacier elevation changes and the fitted elevation changes during 2000-2022.

| Year | 2000 | 2001 | 2002 | 2003 | 2004 | 2005 | 2006 | 2007 | 2008 | 2009 | 2010 | 2011 |
|---|---|---|---|---|---|---|---|---|---|---|---|---|
| Elevation difference | 6.318 | 3.188 | 3.249 | 2.249 | 3.339 | 2.438 | 2.281 | -2.640 | 0.514 | 0.280 | -1.379 | -2.529 |
| Fitted elevation difference | 5.413 | 4.704 | 3.994 | 3.284 | 2.575 | 1.865 | 1.156 | 0.446 | -0.263 | -0.973 | -1.682 | -2.392 |

| Year | 2012 | 2013 | 2014 | 2015 | 2016 | 2017 | 2018 | 2019 | 2020 | 2021 | 2022 | |
|---|---|---|---|---|---|---|---|---|---|---|---|---|
| Elevation difference | -2.434 | -4.608 | -5.254 | -4.742 | -4.171 | -6.141 | -7.631 | -7.295 | -9.547 | -10.563 | -10.624 | |
| Fitted elevation difference | -3.101 | -3.81 | -4.520 | -5.230 | -5.939 | -6.649 | -7.358 | -8.068 | -8.777 | -9.487 | -10.196 | |

*41) 297 'lower' instead of 'reduced'*

R. Thank you for your comment. According to your suggestion, the 'reduced' has been replaced by 'instead' in the revised manuscript.

*42) 300 Say at which elevation the mass balance becomes positive. A graph of this dependence would be highly interesting. Is that valid in the whole study area? Can you compare this with the equilibrium line elevation (ELA)?*

R. Thank you for your comment. The glacier mass change is linearly correlated with the glacier elevation change. Therefore, when the elevation change becomes positive, the mass balance becomes positive. The graph of dependence between the altitude and glacier elevation changes as well as glacier area was provided in our manuscript, and as shown in Fig 2 (b) below, when the altitude was above 5900 m, the glacier elevation change becomes positive, and the mass balance theoretical also becomes positive. The statistics are based on all glaciers in the study area, therefore, our finding in general is valid for the whole study area. Since the equilibrium line elevation (ELA) is the line at which glacier accumulation and ablation are in balance, therefore, the altitude of 5900 m also can be generally regarded as the ELA in the SETP region. The related description has been added to the revised manuscript.

[Figure]

Figure 2(b): the altitudinal distribution of glacier elevation change rates.

***43) 313 This is an unfortunate style of referring to the figures. Why not writing: 'Base on our results (Fig. 3 (a) and (c) it appeared ...'***

R. Thank you for your comment. According to your suggestion, the related sentence have been rephrased in the revised manuscript.

"

4.2 Spatiotemporal variability in glacier elevation change

…

Based on our results (Fig. 3(a) and (c)), it appeared that the fastest glacier thinning events occurred at latitude $29° N$-$29.5° N$ and longitude $97° E$-$97.5° E$

"

***44) Figure 3: The explanation for '(c)' is missing in the caption. Also a legend is needed. 'a' and 'b' are not clear. The caption suggests that there are 3 similar sub-figures. Two are shown instead.***

R. Thank you for your comment. According to your suggestions, the explanation for '(c)' as well as the legend has been added, and the explanations for '(a)' and '(b)' have been rephrased in the revised manuscript.

"

4.2 Spatiotemporal variability in glacier elevation change

…

(a)                                                     (b)

(c)

Figure 3: Illustration of the glacier elevation change during 2000-2022. (a) Map of glacier elevation change rate at a $0.5° \times 0.5°$ grid cell scale for the period of 2000-2022. (b) Map of glacier elevation change rates difference at a $0.5° \times 0.5°$ grid cell scale between the periods of 2000-2011 and 2011-2022. (c) Glacier elevation change rates of the periods of 2000-2011, 2011-2022, and 2000-2022, respectively.

"

***45) 325 The meaning of this chapter is not clear. What the individual arbitrarily selected glacier show about the study region? I guess not much. More meaningful would be to take anomalous glaciers and to analyse those looking at their feeding mechanism, debris cover proportion, orientation, the elevation of the accumulation area or minimal elevation. I suggest to cancel this chapter and deal with the glaciers***

*with positive mass balance and with a sincere analysis of the differences between the various approaches (lidar altimetry, radar altimetry, DEM differencing.*

R. Thank you for your comment. According to your suggestion, we have removed the Chapter 4.3 from our revised manuscript.

According to the derived time-series spatially resolved glacier elevation change from the multisource data, we found that most of the glaciers in SETP show negative elevation change, while there are remain a small portion of glaciers exhibited an anomalous positive elevation change during the period 2000-2022. Then we compared the anomalous glaciers (accumulating glaciers) with the melting glaciers in term of glacier altitude, glacier aspect, glacier number, glacier slope and glacier length, respectively. We found that most of the melting glaciers are facing the northern and eastern sides (North, Northeast, East, Southeast) and most of accumulating glaciers are facing the southwestern side. And the accumulating glaciers characterized in general by a steeper glacier slope, short glacier length, and a slight lager altitude by comparing with the melting glaciers. More details such as the analysis for the feeding mechanism of the accumulating glacier have been provided in the response to your General comments (4) above, which also have been added to the Section 4.2 of the revised manuscript.

We further analyze the differences among the various approaches, such as LiDAR altimetry, Radar altimetry and ASTER DEMs differencing. Firstly, the consistency among different satellite measurements on the glacier region was analyzed, as shown in Fig. 7 (c), the elevation measurements between ICESat and CryoSat-2 were strongly consistent, and the elevations measured by ICESat, ICESat-2 and CryoSat-2 were larger than those of the ASTER DEM. We found that the elevations measured by ICESat-2 were the smallest both in the stable region and in the glacier region. We linked this situation to the spatial resolution of the satellite sensors, specifically, the spatial resolution of the derived ASTER DEM was 30 m, and the footprint diameters of ICESat, ICESat-2 and CryoSat-2 were 70 m, 17.5 m, and 300 m, respectively. Among these satellites, ICESat-2 had the finest spatial resolution. We compared the elevation

measurements of the stable region with those of the glacier region. According to Fig. 7 (b) and (c), the precisions of the elevation measurements obtained by satellite sensors were different between the stable region and glacier region, indicating that the elevation measurements from the ASTER DEM were more consistent with the altimetry data in the stable region; this difference may be mainly due to the relatively flat terrain in the stable region. The related description has been added to the Section 5.1 of the revised manuscript.

[Figure]

Figure 7: Cross-analysis of multisource satellite measurements. The multisource satellite measurements include satellite altimetry of ICESat, ICESat-2, Cryosat-2 and ASTER DEM in this study. The histograms are probability distributions of elevation difference values derived by satellite measurement and DEM image. The colors of histograms are corresponding to different satellite measurements. (a) The elevation difference to be calculated on the stable region by using SRTM DEM, (b) The elevation difference to be calculated on the stable region by using ASTER DEM, and (c) elevation difference to be calculated on the glacier region by using ASTER DEM.

**46) Figure 4: The subfigure (a) should be a separate figure as it does not have much in common with the other sub-figures.**

R. Thank you for your comment. We divided the glaciers into three types based on the glacier elevation change rate, and selected two glaciers for each type for visualization. Specifically, we show the locations of the selected 6 glaciers as well as the linear fitting of elevation changes in subfigure (a), and the elevation difference map corresponding to the selected 6 glaciers are shown in subfigure (b), (c) and (d). Therefore, there are some links between the subfigure (a) and subfigure (b), (c) and (d). According to your suggestion, we have divided this figure into two separate figures, and the related modification has been added the revised manuscript.

"

4.3 Spatiotemporal variability in glacier elevation change

…

[Figure]

Figure 5: Spatial distribution of glacier-wide elevation changes over the study area. The insets show the linear fitting of the elevation changes of the selected glaciers, which corresponding to the glaciers of accumulation type, moderate melting type, and rapid melting type, respectively.

(a) Glacier accumulation

(b) Glacier moderate melting

(c) Glacier rapid melting

[Figure]

Figure 6: Spatial distribution and elevation change rates of glaciers. Elevation difference maps of three-type glaciers of (a) accumulation type, (b) moderate melting type, and (c) rapid melting type, respectively. The spatial distribution and elevation change rates of these glaciers are shown in the Fig.5.

"

**47) The sub-figures show how limiting is to confine the analysis only to the glacier outlines from a global glacier inventory RGI. In the case of a glacier surge, you would not see it at all.**

R. Thank you for your comment. The glaciers which do not flow at a constant speed and to be subjected to cyclical flow instabilities are called 'surging glaciers' or 'surge-type' glaciers. In this study, we investigate the glacier elevation change based on the multisource satellite data and the fixed glacier outline data RGI, Generally, the glacier elevation change occurs in the vertical direction, while the glacier surging occurs mainly in the horizontal direction. In this study we focus on the spatiotemporal analysis of glacier elevation changes and the fixed glacier outline data RGI meets the needs of our study. Nevertheless, due to the lack of time-series accurate glacier outline data, the analysis in terms of glacier surging is limited. Generally, the glacier change includes both the changes from vertical and plane direction, which corresponding to the glacier elevation change and glacier area change. Since the time-series accurate glacier area mapping for the large-scale SETP region is still challenging, therefore, we firstly fully explore the spatiotemporal features of the glacier elevation changes in the SETP and make plans for the dynamic glacier area monitoring as well as the related analysis of glacier surging in the future study.

The related explanation has been added to the Conclusion section of the revised manuscript.

"

6. Conclusion

…

Moreover, the glacier change includes both the changes from vertical and plane direction, which corresponding to the glacier elevation change and glacier area change. we focus on the analysis of spatiotemporal features of the glacier elevation changes, and the glacier area change-related analysis such as snowline variation and glacier surging are lacking in this study. Since the time-series accurate glacier area mapping for the large-scale SETP region is still challenging, therefore, we fully explore the spatiotemporal features of the glacier elevation changes in the SETP and make plans for the dynamic glacier area monitoring as well as the related analysis of glacier surging in the future study.

"

*48) 361 In which way do the Gaussian curves provide information on correlation? I think such a comparison can reveal a systematic error in the methods provided you have a robust reference.*

R. Thank you for your comment. We realize that the expression may not correct here. With your suggestion, we think the systematic error or systematic bias may more appropriate. The "correlation" has been replaced to "system bias" in the corresponding position of the revised manuscript.

*49) Figure 5: This figure is messy. No letters at the sub-figures. Week descriptions in the caption. Provide detailed descriptions of all sub-figures referring to them by a, b and c.*

R. Thank you for your comments. The letters of (a), (b) and (c) have been added to the subfigures of Figure 7 (Figure 5 in the previous manuscript). The caption of Figure

7 has been rephrased, and the captions of subfigures also have been added to Figure 7. The related revision has been added to the Section 5.1 of the revised manuscript.

"

**5.1 Cross-analysis of multisource satellite measurements**

[Figure]

(a)

(b)

(c)

Figure 7: Cross-analysis of multisource satellite measurements. The multisource satellite measurements include satellite altimetry of ICESat, ICESat-2, Cryosat-2 and satellite image-derived ASTER DEM in this study. The histograms are probability distributions of elevation differences derived by satellite measurement and DEM image. The colors of histograms are corresponding to different satellite measurements. (a) The elevation difference to be calculated on the stable region by using SRTM DEM, (b) The elevation difference to be calculated on the stable region by using ASTER DEM, and (c) elevation difference to be calculated on the glacier region by using ASTER DEM.

"

*50) What is meant by 'satellite measurement', is it satellite altimetry by ICESat, Cryosat or something else?*

R. Thank you for your comment. The satellite measurement means the derived elevation by satellite data source. The satellite measurements include the satellite altimetry of ICESat, ICESat-2, Cryosat-2 and the satellite image-derived ASTER DEM in our study. The related description has been added to the explanation of the Figure 7 in the revised manuscript.

***51) 385 The fact, that there is no difference between the strong and weak beams was mentioned in some previous studies. You should cite them here.***

R. Thank you for your comment. The research by Herzfeld et al. (2021) demonstrates that the ICESat-2 data both from strong beam and weak beam can been used to characterize ice surface characteristics, and this means that there is no significant difference between the strong and weak beams in the ice surface measurement. The corresponding paper has been cited, and the related description has been added in the revised manuscript.

**Reference:**

Herzfeld U C, Trantow T, Lawson M, et al. Surface heights and crevasse morphologies of surging and fast-moving glaciers from ICESat-2 laser altimeter data-Application of the density-dimension algorithm (DDA-ice) and evaluation using airborne altimeter and Planet SkySat data[J]. Science of remote sensing, 2021, 3: 100013.

***52) Figure 6: You should explain what you mean by homologous measurements.***

R. Thank you for your comment. The homologous measurements mean the elevation measurements from the same satellite. Due to the difference of acquisition modes or high-level data processing methods, such as the data of strong beam and the weak beam of the ICESat-2, or the L2 (Level 2) SAR data and swatch SARIn data of the CryoSat-2, there may be also differences between the elevation measurements obtained from the same satellite. The related description has been added to the explanation of related figure of the revised manuscript.

**53) Figure 6b: What is the difference between the two means of the distributions? This should be discussed.**

R. Thank you for your comment. The difference between the two means of the distributions illustrates that there is significant systematic bias between the glacier elevation measurements by CryoSat-2 L2 data and CryoSat-2 swath data. By comparing with the CryoSat-2 L2 measurement, the mean of the distribution by CryoSat-2 swath measurements is closer to 0, which illustrates that the CryoSat-2 swath measurement keep higher consistency with the ASTER DEM measurement rather than the CryoSat-2 L2 measurement. Generally, the CryoSat-2 L2 data retrieves surface elevation by determining a Point-Of-Closest-Approach (POCA), meaning that a single elevation is sampled beneath the satellite. CryoSat-2 swath data, however, exploits the full radar waveform to map a dense swath ($\sim$ 5 km wide) of elevation measurements across the satellite ground track (Foresta et al., 2018; Gourmelen et al., 2018),  and providing a 10 to a 100 folds increase in glacier elevation measurement with improved spatial resolution compared with CryoSat-2 L2 data. We compared the performance of CryoSat-2 swath data with CryoSat-2 L2 data in glacier elevation measurement. According to the probability distributions in Figure 8 (b), we found that the CryoSat-2 swath measurement achieved significant improvement in precision by comparing with the CryoSat-2 L2 measurement. The related description and discussion have been added to the revised manuscript.

"

5.1 Cross-analysis of multisource satellite measurements

…

For the CryoSat-2 swath altimetry data, which is mentioned as CryoTEMPO-EOLIS point data in our study, swath processing fully exploits the information contained in CryoSat-2 waveforms and leads to one to two orders of magnitude more measurements

than the CryoSat-2 L2 data processed by the point-of-closest-approach (POCA) technique (Foresta et al., 2018; Gourmelen et al., 2018).

…

CryoSat-2 swath and L2 measurements were compared on the glacier region. According to Fig. 8 (b), There is significant difference between the means of the probability distributions of CryoSat-2 swath and L2 measurements. The difference illustrates that there is a significant systematic bias between the glacier elevation measurements by CryoSat-2 swath data and CryoSat-2 L2 data. By comparing with the CryoSat-2 L2 measurement, the mean of the distribution by CryoSat-2 swath measurements is closer to 0, which illustrates that the CryoSat-2 swath measurement is consistent with the ASTER DEM measurement, while the CryoSat-2 L2 measurements showed large differences from the ASTER DEM measurements. According to the cross validation with ASTER DEM data, we can conclude that the CryoSat-2 swath data achieved significant improvement in precision by comparing with the CryoSat-2 L2 data in glacier elevation measurements."

**Reference:**

Foresta, L., Gourmelen, N., Pálsson, F., Nienow, P., Björnsson, H., and Shepherd, A.: Surface elevation change and mass balance of Icelandic ice caps derived from swath mode CryoSat-2 altimetry, Geophys. Res. Lett., 43, 12138–12145

Gourmelen, N., Escorihuela, M., Shepherd, A., Foresta, L., Muir, A., Garcia-Mondejar, A., Roca, M., Baker, S., and Drinkwater, M. R.: CryoSat-2 swath interferometric altimetry for mapping ice elevation and elevation change, Adv. Space Res., 62, 1226-1242.

*54) Figure 7: This is a highly important figure. The sub-figure a should be larger and the lines should be easier to distinguish, for instance by a small offset if all start on the same date. If the figure is large enough you might add the info on the instrument to (a) and remove (b). Why is actually the y-axis inverse? Is there any strong reason to have the negative values above?*

R. Thank you for your comment. According to your suggestion, we have made a small offset for the glacier elevation change estimates with the same start time. And we have added the data source information to subfigure (a) and the subfigure (b) is then removed from the Figure 9 (Figure 7 in our previous manuscript).

The reason to make the y-axis inversed is that 1) the elevation change rates in SETP are negative in all the exiting research and 2) we want to make the y-axis start at 0. According to your suggestion, we have modified the y-axis name of "Elevation change rate (m/yr)" to "Negative elevation change rate (-m/yr)", thus the y-axis to be set as usual and the value above is larger than the value below.

The related revision has been added to the Section of 5.2 in the revised manuscript.

"

**5.2 Comparison of existing estimates of glacier elevation change**

…

[Figure]

Figure 9: Estimates of glacier elevation change rate in published studies. The dashed box represents the uncertainty of the estimated glacier elevation change, and we labeled the data source for the glacier elevation change estimation in the figure.

"

Once again, thank you for your time reviewing this paper.

Best wishes,

The authors